# Online Bayesian Persuasion

**Matteo Castiglioni**
Politecnico di Milano
matteo.castiglioni@polimi.it

**Andrea Celli**[*]
Facebook Core Data Science
andreacelli@fb.com

**Alberto Marchesi**
Politecnico di Milano
alberto.marchesi@polimi.it

**Nicola Gatti**
Politecnico di Milano
nicola.gatti@polimi.it

## Abstract

In Bayesian persuasion, an informed sender has to design a signaling scheme that discloses the right amount of information so as to influence the behavior of a self-interested receiver. This kind of strategic interaction is ubiquitous in real-world economic scenarios. However, the seminal model by Kamenica and Gentzkow makes some stringent assumptions that limit its applicability in practice. One of the most limiting assumptions is, arguably, that the sender is required to know the receiver's utility function to compute an optimal signaling scheme. We relax this assumption through an *online learning* framework in which the sender repeatedly faces a receiver whose type is unknown and chosen adversarially at each round from a finite set of possible types. We are interested in *no-regret* algorithms prescribing a signaling scheme at each round of the repeated interaction with performances close to that of a best-in-hindsight signaling scheme. First, we prove a hardness result on the per-round running time required to achieve no-$\alpha$-regret for any $\alpha < 1$. Then, we provide algorithms for the *full* and *partial feedback* models with regret bounds sublinear in the number of rounds and polynomial in the size of the instance.

## 1 Introduction

Bayesian persuasion was first introduced by Kamenica and Gentzkow [23] as the problem faced by an informed *sender* trying to influence the behavior of a self-interested *receiver* via the strategic provision of payoff-relevant information. In Bayesian persuasion, the agents' beliefs are influenced only by controlling 'who gets to know what'. This 'sweet talk' is ubiquitous among all sorts of economic activities, and it was famously attributed to a quarter of the GDP in the United States by McCloskey and Klamer [28]. [2] The computational study of Bayesian persuasion has been largely driven by its application in domains such as auctions and online advertisement [7, 19, 11], voting [1, 14, 16], traffic routing [9, 32], recommendation systems [26], security [30, 34], and product marketing [6, 13].

In the model by Kamenica and Gentzkow [23], the sender's and receiver's payoffs are determined by the receiver's action and a set of parameters collectively termed the *state of nature*. Unlike the receiver, the sender observes the realized state of nature drawn from a shared prior distribution. The sender uses this private information to determine a signal for the receiver according to a publicly known *signaling scheme*, *i.e.*, a mapping from states of nature to probability distributions over signals.

In this paper, we focus on arguably one of the most severe limitations of the basic model: the sender must know exactly the receiver's utility function to compute an optimal signaling scheme.

---

[*]The work was conducted while the author was a postdoc at Politecnico di Milano.
[2]A more recent estimate by Antioch and others [2] places this figure at 30%.

**Our model and results**  We deal with uncertainty about the receiver's type by framing the Bayesian persuasion problem in an online learning framework. We study a repeated Bayesian persuasion problem where, at each round, the receiver's type is adversarially chosen from a finite set of types. Our goal is the design of an online algorithm that recommends a signaling scheme at each round, guaranteeing an expected utility for the sender close to that of the best-in-hindsight signaling scheme. We study this problem under two models of feedback: in the *full information* model, the sender selects a signaling scheme and later observes the type of the best-responding receiver; in the *partial information* model, the sender only observes the actions taken by the receiver.

First, in Section 4, we provide a negative result that rules out, even in the full information setting, the possibility of designing a no-regret algorithm with polynomial per-round running time. Furthermore, the same hardness result holds when adopting the notion of no-$\alpha$-regret (in the additive sense) for any $\alpha < 1$. Then, we focus on the problem of designing no-regret algorithms by relaxing the running time constraint. We show that it is possible to achieve a regret polynomial in the size of the problem instance and sublinear in the number of rounds $T$ under both full (with $O(T^{-1/2})$) and partial feedback (with $O(T^{-1/5})$). We present these results in Sections 5 and 6, respectively.

**Related works**  The closest line of research to ours is the one studying online learning problems in *Stackelberg games*. In these games, a *leader* commits to a probability distribution over a set of actions, and a *follower* plays an action maximizing her/his utility given the leader's commitment [33]. In this setting, Letchford *et al.* [25] and Blum *et al.* [10] study the problem of computing the best leader's strategy against an unknown follower using a polynomial number of best-response queries. Marecki *et al.* [27] study the problem with a single follower with type drawn from a Bayesian prior.

Balcan *et al.* [8] study how to minimize the leader's regret in an online setting in which the follower's type is unknown and chosen adversarially from a finite set. Although the problem is conceptually similar to ours, the Bayesian persuasion framework presents a number of additional challenges: the solution to a Stackelberg game consists of a point in a finite-dimensional simplex, while the solution to a Bayesian persuasion problem is a probability distribution with potentially infinite support size. This probability distribution is subject to additional consistency constraints, which (under partial feedback) rule out the possibility of exploiting unbiased estimators of the sender's expected utility.

Finally, it is worth mentioning that known online learning algorithms (for either the full or partial feedback setting) do not provide any guarantee in the case of Bayesian persuasion. Indeed, the regret bounds of those algorithms depend linearly or sublinearly in the number of actions, but the action space in Bayesian persuasion is infinite. A large body of previous works in other fields resolves the issue of dealing with an infinite action space by requiring specific assumptions (*e.g.*, linear or convex utility function [4, 12, 22, 35]). However, in the online Bayesian persuasion setting, these assumptions do not hold as the sender's utility depends on the receiver's best response, which yields a function that is not linear nor convex (or even continuous in the space of signaling schemes).

## 2  Preliminaries

The receiver has a finite set of $m$ actions $\mathcal{A} := \{a_i\}_{i=1}^m$ and a set of $n$ possible types $\mathcal{K} := \{k_i\}_{i=1}^n$. For each type $k \in \mathcal{K}$, the receiver's payoff function is $u^k : \mathcal{A} \times \Theta \to [0, 1]$, where $\Theta := \{\theta_i\}_{i=1}^d$ is a finite set of $d$ states of nature. For notational convenience, we denote by $u_\theta^k(a) \in [0, 1]$ the utility observed by the receiver of type $k \in \mathcal{K}$ when the realized state of nature is $\theta \in \Theta$ and she/he plays action $a \in \mathcal{A}$. The sender's utility when the state of nature is $\theta \in \Theta$ is described by the function $u_\theta^\mathsf{s} : \mathcal{A} \to [0, 1]$. As it is customary in Bayesian persuasion, we assume that the state of nature is drawn from a common prior distribution $\boldsymbol{\mu} \in \text{int}(\Delta_\Theta)$, which is explicitly known to both the sender and the receiver. [3] Moreover, the sender can commit to a *signaling scheme* $\phi$, which is a randomized mapping from states of nature to *signals* for the receiver. Formally $\phi : \Theta \to \Delta_\mathcal{S}$, where $\mathcal{S}$ is a finite set of signals. We denote by $\phi_\theta$ the probability distribution employed by the sender when the state of nature is $\theta \in \Theta$, with $\phi_\theta(s)$ being the probability of sending signal $s \in \mathcal{S}$.

A *one-shot* interaction between the sender and the receiver goes on as follows: (i) the sender commits to a publicly known signaling scheme $\phi$ and the receiver observes the commitment; (ii) the sender

observes the realized state of nature $\theta \sim \boldsymbol{\mu}$; (iii) the sender draws a signal $s \sim \phi_\theta$ and communicates it to the receiver; (iv) the receiver observes $s$ and rationally updates her/his prior beliefs over $\Theta$ according to the *Bayes rule*; (v) the receiver selects an action maximizing her/his expected utility.

Let $\Xi := \Delta_\Theta$ be the set of receiver's posterior beliefs over the states of nature. In step (iv), after observing $s \in \mathcal{S}$, the receiver performs a Bayesian update and infers a posterior belief $\boldsymbol{\xi} \in \Xi$ over the states of nature such that the component of $\boldsymbol{\xi}$ corresponding to state of nature $\theta \in \Theta$ is:

$$\xi_\theta := \frac{\mu_\theta \, \phi_\theta(s)}{\sum_{\theta' \in \Theta} \mu_{\theta'} \, \phi_{\theta'}(s)}. \tag{1}$$

After computing $\boldsymbol{\xi}$, the receiver solves a decision problem to find an action maximizing her/his expected utility given the current posterior. Letting $a \in \mathcal{A}$ be the receiver's choice, the receiver observes payoff $u_\theta^k(a)$, where $k \in \mathcal{K}$ is the receiver's type, while the sender observes payoff $u_\theta^s(a)$.

## 2.1 Working in the space of posterior distributions

It is oftentimes useful to represent signaling schemes as convex combinations of posterior beliefs they can induce. First, we describe such interpretation (see [24] for further details). Then, we define the receiver's best response given an arbitrary posterior belief.

**Representing signaling schemes** Given a signaling scheme $\phi$, each signal realization $s \in \mathcal{S}$ leads to a posterior belief $\boldsymbol{\xi}^s \in \Xi$, whose components are defined as in Equation (1). Accordingly, each signaling scheme leads to a distribution over posterior beliefs. We denote a distribution over posteriors by $\mathbf{w} \in \Delta_\Xi$. We say that a signaling scheme $\phi : \Theta \to \Delta_\mathcal{S}$ *induces* $\mathbf{w} \in \Delta_\Xi$ if, for every $\boldsymbol{\xi} \in \Xi$, the component of $\mathbf{w}$ corresponding to $\boldsymbol{\xi}$ is defined as follows:

$$w_{\boldsymbol{\xi}} := \sum_{s \in \mathcal{S} : \boldsymbol{\xi}^s = \boldsymbol{\xi}} \sum_{\theta \in \Theta} \mu_\theta \, \phi_\theta(s). \tag{2}$$

Intuitively, if $\phi$ induces $\mathbf{w}$, then $w_{\boldsymbol{\xi}}$ represents the probability that $\phi$ induces the posterior $\boldsymbol{\xi} \in \Xi$. We let $\mathrm{supp}(\mathbf{w}) := \{\boldsymbol{\xi} \in \Xi \mid w_{\boldsymbol{\xi}} > 0\}$ be the set of posteriors induced with strictly positive probability. We say that a distribution over posteriors $\mathbf{w} \in \Delta_\Xi$ is *consistent* (*i.e.*, intuitively, there exists a valid signaling scheme $\phi$ inducing $\mathbf{w}$) if the following hods:

$$\sum_{\boldsymbol{\xi} \in \mathrm{supp}(\mathbf{w})} w_{\boldsymbol{\xi}} \, \xi_\theta = \mu_\theta, \quad \text{for all } \theta \in \Theta. \tag{3}$$

We let $W \subseteq \Delta_\Xi$ be the set of distributions over posteriors that are consistent according to Equation (3). In the remainder of the paper, we equivalently employ $\phi$ or $\mathbf{w}$ to denote an arbitrary signaling scheme.

**Receiver's best-response set** After observing a signal $s \in \mathcal{S}$ that induces a posterior $\boldsymbol{\xi} \in \Xi$, the receiver best responds by choosing an action that maximizes her/his expected utility (step (v)). The set of actions maximizing the receiver's expected utility given posterior $\boldsymbol{\xi}$ is defined as follows:

**Definition 1** (BR-set). *Given posterior $\boldsymbol{\xi} \in \Xi$ and type $k \in \mathcal{K}$, the* best-response set (BR-set) *is:*

$$\mathcal{B}_{\boldsymbol{\xi}}^k := \arg\max_{a \in \mathcal{A}} \sum_{\theta \in \Theta} \xi_\theta \, u_\theta^k(a).$$

We denote by $b_{\boldsymbol{\xi}}^k$ the action belonging to the BR-set $\mathcal{B}_{\boldsymbol{\xi}}^k$ played by the receiver. When the receiver is indifferent among multiple actions for a given posterior $\boldsymbol{\xi}$, we assume that the receiver breaks ties in favor of the sender, *i.e.*, she/he chooses an action $b_{\boldsymbol{\xi}}^k \in \arg\max_{a \in \mathcal{B}_{\boldsymbol{\xi}}^k} \sum_\theta \xi_\theta \, u_\theta^s(a)$. [4]

We conclude the section by introducing some additional notation. We denote by $u^s(\boldsymbol{\xi}, k) := \sum_\theta \xi_\theta \, u_\theta^s(b_{\boldsymbol{\xi}}^k)$ the sender's expected utility when she/he induces a posterior $\boldsymbol{\xi} \in \Xi$ and the receiver is of type $k \in \mathcal{K}$. Moreover, we use $u^s(\phi, k)$ to denote the sender's expected utility achieved with the signaling scheme $\phi$. Formally, $u^s(\phi, k) := \sum_{\boldsymbol{\xi} \in \mathrm{supp}(\mathbf{w})} w_{\boldsymbol{\xi}} \, u^s(\boldsymbol{\xi}, k)$, where $\mathbf{w} \in \Delta_\Xi$ is the distribution over posteriors induced by $\phi$. Analogously, we write $u^s(\mathbf{w}, k)$.

Finally, letting $OPT$ be the sender's optimal expected utility, we say that a signaling scheme is *$\alpha$-optimal* (in the additive sense) if it provides the sender with a utility at least as large as $OPT - \alpha$.

| $\mathcal{A}$ | | State G $(\mu_{\mathrm{G}} = .3)$ | | State I $(\mu_{\mathrm{I}} = .7)$ | |
|---|---|---|---|---|---|
| A | $\ \|$ | 0 | 0 | 0 | 1 |
| C | | 1 | 1 | 1 | 0 |

| $\mathcal{S}$ | | Realized state | |
|---|---|---|---|
| | | State G | State I |
| $s_1$ | $\ \|$ | 0 | 4/7 |
| $s_2$ | | 1 | 3/7 |

| $\mathrm{supp}(\mathbf{w}^\star)$ | | State of nature | | $\mathbf{w}^\star$ |
|---|---|---|---|---|
| | | State G | State I | |
| $\boldsymbol{\xi}_1$ | $\ \|$ | 0 | 1 | 2/5 |
| $\boldsymbol{\xi}_2$ | | 1/2 | 1/2 | 3/5 |

Figure 1: **Left:** The prosecutor/judge game. Rows represent the judge's actions. For each possible state of nature $\{\mathrm{G}, \mathrm{I}\}$, the first column is the prosecutor's payoff while the second is the judge's payoff. **Center:** The optimal signaling scheme $\phi^\star$. Each column describes the probability with which the two signals are drawn given the realized state of nature. **Right**: Representation of $\phi^\star$ as the convex combination of posteriors $\mathbf{w}^\star$.

## 2.2 Example

We illustrate the key notion of signaling scheme in a simple example with a single receiver type (*i.e.,* $|\mathcal{K}| = 1$) inspired by Kamenica and Gentzkow [23]: a prosecutor (the sender) is trying to convince a rational judge (the receiver) that a defendant is guilty. The judge has two available actions: to *acquit* or to *convict* the defendant (denoted by A and C, respectively). There are two possible states of nature: the defendant is either *guilty* (denoted by G) or *innocent* (denoted by I). The prosecutor and the judge share a prior belief $\mu_{\mathrm{G}} = .3$. Moreover, the prosecutor gets utility 1 if the judge convicts the defendant and 0 otherwise, regardless of the state of nature. The prosecutor gets to observe the realized state of nature (*i.e.,* whether the defendant is guilty or innocent). The she/he can exploit this information to select a signal from set $\mathcal{S} = \{s_1, s_2\}$ and send it to the judge. The judge has a unique type and she/he gets utility 1 for choosing the just action (convict when guilty and acquit when innocent) and utility 0 for choosing the unjust action (see Figure 1-Left).

Figure 1-Center depicts a sender-optimal signaling scheme $\phi^\star$ obtained via the following LP:

$$\underset{\phi \geq 0}{\arg\max}\, u^{\mathsf{s}}(\phi, k) \quad \text{s.t.} \quad \sum_{s \in \mathcal{S}} \phi_\theta(s) = 1 \quad \forall \theta \in \Theta,$$

where $k$ is the unique type of the judge. When the sender acts according to $\phi^\star$, signal $s_1$ (resp., $s_2$) originates posterior $\boldsymbol{\xi}_1$ (resp., $\boldsymbol{\xi}_2$; see Figure 1-Right). Applying Equation (3) yields the equivalent representation of $\phi^\star$ as a convex combination of posteriors, *i.e.*, $w^\star_{\boldsymbol{\xi}_1} = 2/5$ and $w^\star_{\boldsymbol{\xi}_2} = 3/5$.

By unpacking the objective function of the above LP (and dropping the dependency on $k$) we have: $\mathcal{B}_{\boldsymbol{\xi}_1} = \{\mathrm{A}\}$ and $\mathcal{B}_{\boldsymbol{\xi}_2} = \{\mathrm{A}, \mathrm{C}\}$. Therefore, if the posterior is $\boldsymbol{\xi}_1$, the judge will acquit the defendant, *i.e.,* $b_{\boldsymbol{\xi}_1} = \mathrm{A}$. Otherwise, if the posterior is $\boldsymbol{\xi}_2$, we have $b_{\boldsymbol{\xi}_2} = \mathrm{C}$ since the receiver breaks ties in favor of the sender. This highlights an intuitive interpretation of the signaling problem: the two signals may be interpreted as action recommendations. Signal $s_1$ (resp., $s_2$) is interpreted by the judge as a recommendation to play A (resp., C). Then, our definition of best-response set (Definition 1) implies that it is in the receiver's best interest to follow the action recommendations. The best-response conditions can be formulated in terms of linear constraints on $\phi_\theta$ as follows:

$$\sum_{\theta \in \Theta} \mu_\theta\, \phi_\theta(s_1) \Big( u_\theta(\mathrm{A}) - u_\theta(\hat{a}) \Big) \geq 0 \quad \text{and} \quad \sum_{\theta \in \Theta} \mu_\theta\, \phi_\theta(s_2) \Big( u_\theta(\mathrm{C}) - u_\theta(\hat{a}) \Big) \geq 0 \quad \forall \hat{a} \in \{\mathrm{A}, \mathrm{C}\}.$$

## 3 The online Bayesian persuasion framework

We consider the following online setting. The sender plays a repeated game in which, at each round $t \in [T]$, she/he commits to a signaling scheme $\phi^t$, observes a state of nature $\theta^t \sim \boldsymbol{\mu}$, and she/he sends signal $s^t \sim \phi^t_{\theta^t}$ to the receiver. [5] Then, a receiver of unknown type updates her/his prior distribution and selects an action $a^t$ maximizing her/his expected reward (in the *one-shot* interaction at round $t$). We focus on the problem in which the sequence of receiver's types $\mathbf{k} := \{k^t\}_{t \in [T]}$ is selected beforehand by an adversary. After the receiver plays $a^t$, the sender receives a *feedback* on her/his choice at round $t$. In the *full information* feedback setting, the sender observes the receiver's type $k^t$. Therefore, the sender can compute the expected payoff for any signaling scheme she/he could have chosen other than $\phi^t$. Instead, in the *partial information* feedback setting, the sender only observes the action $a^t$ played by the receiver at round $t$.

We are interested in algorithms computing $\phi^t$ at each round $t$. The performance of one such algorithm is measured using the average per-round *regret* computed with respect to the best signaling scheme in hindsight. Formally:

$$R^T := \max_{\phi} \left\{ \frac{1}{T} \sum_{t=1}^{T} \left( u^{\mathsf{s}}(\phi, k^t) - \mathbb{E}\left[ u^{\mathsf{s}}(\phi^t, k^t) \right] \right) \right\},$$

where the expectation is on the randomness of the online algorithm (*i.e.*, the probability distribution which is used by the sender to draw the signaling scheme at round $t$) and $T$ is the number of rounds. Ideally, we would like to find an algorithm that generates a sequence $\{\phi^t\}_{t \in [T]}$ with the following properties: (i) the regret is polynomial in the size of the problem instance, *i.e.*, $\mathsf{poly}(n, m, d)$, and goes to zero as a polynomial of $T$; (ii) the per-round running time is $\mathsf{poly}(t, n, m, d)$. An algorithm satisfying property (i) is usually called a *no-regret* algorithm.

In the case in which requiring no-regret is too limiting, we use the following relaxed notion of regret. An algorithm has *no-$\alpha$-regret* if there exists a constant $c > 0$ such that: $R^T \leq \alpha + \frac{1}{T^c} \mathsf{poly}(n, m, d)$. The idea of no-$\alpha$-regret is that the regret approaches $\alpha$ after a sufficiently large number of rounds (polynomial in the size of the game).

## 4 Hardness of sub-linear regret

Our first result is negative: for any $\alpha < 1$, it is unlikely (*i.e.*, technically, it is not the case unless $\mathsf{NP} \subseteq \mathsf{RP}$) that there exists a no-$\alpha$-regret algorithm for the online Bayesian persuasion problem requiring a per-round running time polynomial in the size of the instance. In order to prove the result, we provide an intermediate step, showing that the problem of approximating an optimal signaling scheme is computationally intractable even in the *offline* Bayesian persuasion problem in which the sender knows the probability distribution over the receiver's types (see Theorem 1 below).

**Definition 2** (OPT-SIGNAL). *Given an offline Bayesian persuasion problem in which the distribution over the receiver's types $\rho \in \Delta_{\mathcal{K}}$ is uniform, i.e., $\rho_k = \frac{1}{n}$ for all $k \in \mathcal{K}$, we call OPT-SIGNAL the problem of finding an optimal signaling scheme $\phi : \Theta \to \Delta_{\mathcal{S}}$, i.e., one maximizing the sender's expected utility $\frac{1}{n} \sum_{k \in \mathcal{K}} u^{\mathsf{s}}(\phi, k)$.*

Then, we can prove the following result (the omitted proofs can be found in Appendix B).

**Theorem 1.** *For every $0 \leq \alpha < 1$, it is NP-hard to compute an $\alpha$-optimal solution to OPT-SIGNAL.*

Now, we use the approximation-hardness of the offline Bayesian persuasion problem to provide lower bounds on the $\alpha$-regret in the online setting. In order to do this, we employ a set of techniques introduced by Roughgarden and Wang [31], which lead to the following result. [6].

**Theorem 2.** *For every $\alpha < 1$, there is no polynomial-time algorithm for the online Bayesian persuasion problem providing no-$\alpha$-regret, unless $\mathsf{NP} \subseteq \mathsf{RP}$.*

## 5 Full information feedback setting

The negative result of the previous section (Theorem 2) rules out the possibility of designing an algorithm which satisfies the no-regret property and requires a $\mathsf{poly}(t, n, m, d)$ per-round running time. A natural question is whether it is possible to devise a no-regret algorithm for the online Bayesian persuasion problem by relaxing the running-time constraint. This is not a trivial problem because, at every round $t$, the sender has to choose a signaling scheme among an infinite number of alternatives and her/his utility depends on the receiver's best response, which yields a function that is not linear nor convex (or even continuous in the space of the signaling schemes). However, we show that it is possible to provide a no-regret algorithm for the full information setting by restricting the sender's action space to a finite set of posteriors. All the omitted proofs are in Appendix C.

First, we show that it is always possible to design a sender-optimal signaling scheme defined as a convex combination of a specific finite set of posteriors. For each type $k \in \mathcal{K}$ and action $a \in \mathcal{A}$, we define $\Xi_a^k \subseteq \Delta_{\Theta}$ as the set of posterior beliefs in which $a$ is a receiver's best response. Formally,

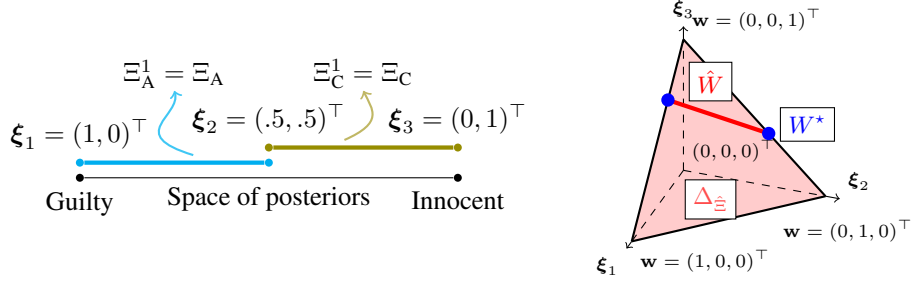

Figure 2: *Left*: Subdivision of the space of posteriors $\Xi$ in the two best-response regions. If $\boldsymbol{\xi} \in \Xi_A$ (resp., $\boldsymbol{\xi} \in \Xi_C$) then the judge's best response under $\boldsymbol{\xi}$ is acquitting (resp., convicting) the defendant. When $\boldsymbol{\xi} = \boldsymbol{\xi}_2$, the judge is indifferent among her/his available actions. We have $\hat{\Xi} = \{\boldsymbol{\xi}_1, \boldsymbol{\xi}_2, \boldsymbol{\xi}_3\}$. *Right*: Visual depiction of $\Delta_{\hat{\Xi}}$, $\hat{W} \subseteq \Delta_{\hat{\Xi}}$, and $W^\star = V(\hat{W})$. The set $\hat{W}$ comprises of the distributions over posteriors in $\hat{\Xi}$ consistent with the prior $\boldsymbol{\mu} = (.3, .7)^\top$ and it is obtained by intersecting $\Delta_{\hat{\Xi}}$ with $[\boldsymbol{\xi}_1 \mid \boldsymbol{\xi}_2 \mid \boldsymbol{\xi}_3] \cdot \mathbf{w} \geq \boldsymbol{\mu}$. As a result, we obtain $\hat{W} = \text{conv}\{(.3, 0, .7)^\top, (0, .6, .4)^\top\}$. Finally, $W^\star = V(\hat{W}) = \{(.3, 0, .7)^\top, (0, .6, .4)^\top\}$.

$\Xi_a^k := \left\{\boldsymbol{\xi} \in \Xi \mid a \in \mathcal{B}_{\boldsymbol{\xi}}^k\right\}$. Let $\mathbf{a} = (a^k)_{k \in \mathcal{K}} \in \bigtimes_{k \in \mathcal{K}} \mathcal{A}$ be a tuple specifying one action for each receiver's type $k$. Then, for each tuple $\mathbf{a}$, let $\Xi_{\mathbf{a}} \subseteq \Delta_\Theta$ be the (potentially empty) polytope such that each action $a^k$ is optimal for the corresponding type $k$, *i.e.,* $\Xi_{\mathbf{a}} := \bigcap_{k \in \mathcal{K}} \Xi_{a^k}^k$. The polytope $\Xi_{\mathbf{a}}$ has a simple interpretation: a probability distribution over posteriors in $\Xi_{\mathbf{a}}$ yields a signaling scheme such that, for every type $k$, the receiver has no interest in deviating from $a^k$ in the induced posteriors $\Xi_{\mathbf{a}}$ (*i.e.,* the constraints analogous to those of the example in Section 2.2 are satisfied).

Then, let $\hat{\Xi} \subseteq \Xi$ be the set of posteriors defined as $\hat{\Xi} := \bigcup_{\mathbf{a} \in \bigtimes_{k \in \mathcal{K}} \mathcal{A}} V(\Xi_{\mathbf{a}})$. [7] Finally, we define the following set of consistent (according to Equation (3)) distributions over posteriors in $\hat{\Xi}$:

$$\hat{W} := \left\{\mathbf{w} \in \Delta_{\hat{\Xi}} \mid \sum_{\boldsymbol{\xi} \in \hat{\Xi}} w_\theta \xi_\theta = \mu_\theta, \ \forall \theta \in \Theta\right\}. \tag{4}$$

By letting $M$ be a suitably defined $|\Theta| \times |\hat{\Xi}|$-dimensional matrix with one column for each $\xi \in \hat{\Xi}$, then the affine hyperplanes defined by Equation (3) are in the form $M \cdot \mathbf{w} = \boldsymbol{\mu}$. Since $\mathbf{w} \in \Delta_{\hat{\Xi}}$, we can safely rewrite the consistency constraints as $M \cdot \mathbf{w} \geq \boldsymbol{\mu}$ (see the example below for a better intuition). Then, $\hat{W}$ can be seen as the intersection between the simplex $\Delta_{\hat{\Xi}}$ and a finite number of half-spaces. Therefore, $\hat{W}$ is a convex polytope, whose vertices compose the finite action space that will be employed by the no-regret algorithm. Specifically, let

$$W^\star := V(\hat{W}). \tag{5}$$

**Example** Consider the game of Section 2.2 (see Figure 1–Left) where the receiver has a single type (*type 1*). We obtain $\hat{\Xi}$ by partitioning the space of posteriors in different best response regions and by taking the vertices of the resulting polytopes (see Figure 2–Left). Then, we provide a visual depiction of $\hat{W}$ and $W^\star$, which are obtained, respectively, by intersecting $\Delta_{\hat{\Xi}}$ with the hyperplanes corresponding to consistency constraints (see Equation (4)), and then taking the vertices of the resulting polytope (see Figure 2–Right). Another example, with two receiver's types, is provided in Appendix A.

For an arbitrary sequence of receiver's types, we show that there exists $\mathbf{w}^\star \in W^\star$ guaranteeing to the sender an expected utility that is equal to the best-in-hindsight signaling scheme.

**Lemma 1.** *For every sequence of receiver's types $\boldsymbol{k} = \{k^t\}_{t \in [T]}$, it holds*

$$\max_{\mathbf{w} \in W} \sum_{t=1}^{T} u^s(\mathbf{w}, k^t) = \max_{\mathbf{w}^\star \in W^\star} \sum_{t=1}^{T} u^s(\mathbf{w}^\star, k^t).$$

The size of the sender's finite action space grows exponentially in the number of states of nature $d$.

**Lemma 2.** *The size of $W^\star$ is $|W^\star| \in O\left((n\,m^2 + d)^d\right)$.*

Now, by letting $\eta \in [0,1]$ be the maximum absolute payoff value, we can employ any algorithm satisfying $R^T \leq O\left(\eta\sqrt{\log|A|/T}\right)$ as a black box (see, *e.g.*, *Polynomial Weights* [15] and *Follow the Lazy Leader* [22]). By taking $W^\star$ as the sender action space, we obtain the following.

**Theorem 3.** *Given an online Bayesian persuasion problem with full information feedback, there exists an online algorithm such that, for every sequence of receiver's types $\boldsymbol{k} = \{k^t\}_{t \in [T]}$:*

$$R^T \leq O\left(\sqrt{\frac{d\,\log(n\,m^2 + d)}{T}}\right).$$

Notice that any no-regret algorithm working on $W^\star$ requires a per-round running time polynomial in $n, m$ and exponential in $d$ (see the bound in Lemma 2). This shows that the source of the hardness result in Theorem 2 is the number of states of nature $d$, while achieving no-regret in polynomial time is possible when the parameter $d$ is fixed.

## 6 Partial information feedback setting

In this setting, at every round $t$, the sender can only observe the action $a^t$ played by the receiver. Therefore, the sender has no information on the utility $u^s(\mathbf{w}, k^t)$ that she/he would have obtained by choosing any signaling scheme $\mathbf{w} \in W^\star$ other than $\mathbf{w}^t$. We show how to design no-regret algorithms with regret bounds that depend polynomially in the size of the problem instance by exploiting a reduction from the partial information setting to the full information one. [8] The main idea is to use a full-information no-regret algorithm in combination with a mechanism to estimate the sender's utilities corresponding to signaling schemes different from the one recommended by the algorithm. In particular, the overall time horizon $T$ is split into a given number of equally-sized blocks, each corresponding to a window of time simulating a single round of a full information setting. During this window, the strategy suggested by the full-information algorithm is played in most of the rounds (exploitation phase), while only few rounds are chosen uniformly at random and used by the mechanism that estimates the utilities provided by other signaling schemes (exploration phase). Algorithm 1 provides a sketch of the overall procedure, where $Z$ (Line 1) denotes the number of blocks, which are the intervals of consecutive rounds $\{I_\tau\}_{\tau \in [Z]}$ defined in Line 4. The FULL-INFORMATION$(\cdot)$ sub-procedure is a black box representing a no-regret algorithm for the full information setting, working on a subset $W^\circ \subseteq W^\star$ of signaling schemes. After the execution of all the rounds of each block $\tau \in [Z]$, it takes as input the utility estimates computed during $I_\tau$ and returns a recommended strategy $\mathbf{q}^{\tau+1} \in \Delta_{W^\circ}$ for the next block $I_{\tau+1}$ (see Line 14).

---

**Algorithm 1** ONLINE BAYESIAN PERSUASION WITH PARTIAL INFORMATION FEEDBACK

---

**Input:** Full-information no-regret algorithm FULL-INFORMATION$(\cdot)$ working on $W^\circ \subseteq W^\star$; subset of signaling schemes $W^\circledcirc \subseteq W^\star$ used for exploration $\quad \triangleright$ See Appendix D.2 for the definitions of $W^\circ$ and $W^\circledcirc$
1: Let $Z$ be defined as in Theorem 3
2: Let $\mathbf{q}^1 \in \Delta_{W^\circ}$ be the uniform distribution over $W^\circ$
3: **for** $\tau = 1, \ldots, Z$ **do**
4: $\quad I_\tau \leftarrow \left\{(\tau - 1)\frac{T}{Z} + 1, \ldots, \tau\frac{T}{Z}\right\}$
5: $\quad$ Choose a random permutation $\pi : [|W^\circledcirc|] \to W^\circledcirc$ and $t_1, \ldots, t_{|W^\circledcirc|}$ rounds at random from $I_\tau$
6: $\quad$ **for** $t = (\tau - 1)\frac{T}{Z} + 1, \ldots, \tau\frac{T}{Z}$ **do**
7: $\quad\quad$ **if** $t = t_j$ for some $j \in [|W^\circledcirc|]$ **then**
8: $\quad\quad\quad \mathbf{q}^t \leftarrow \mathbf{q} \in \Delta_{W^\star}$ such that $q_\mathbf{w} = 1$ for the signaling scheme $\mathbf{w} = \pi(j)$ $\quad \triangleright$ Exploration phase
9: $\quad\quad$ **else**
10: $\quad\quad\quad \mathbf{q}^t \leftarrow \mathbf{q}^\tau$ $\quad\quad\quad\quad\quad\quad\quad\quad\quad\quad\quad\quad\quad\quad\quad\quad\quad\quad\quad \triangleright$ Exploitation phase
11: $\quad$ Play a signaling scheme $\mathbf{w}^t \in W^\star$ randomly drawn from $\mathbf{q}^t$
12: $\quad$ Observe sender's utility $u^s(\mathbf{w}^t, k^t)$ and receiver's action $a^t \in \mathcal{A}$
13: $\quad$ Compute estimators $\tilde{u}^s_{I_\tau}(\mathbf{w})$ of $u^s_{I_\tau}(\mathbf{w}) \coloneqq \frac{1}{|I_\tau|}\sum_{t \in [T]:t \in I} u^s(\mathbf{w}, k^t)$ for all $\mathbf{w} \in W^\circ$
14: $\quad \mathbf{q}^{\tau+1} \leftarrow$ FULL-INFORMATION$\left(\left\{\tilde{u}^s_{I_\tau}(\mathbf{w})\right\}_{\mathbf{w} \in W^\circ}\right)$

During each block $I_\tau$ with $\tau \in [Z]$, Algorithm 1 alternates between two tasks: (i) *exploration* (Line 8), trying all the signaling schemes in a subset $W^{\circledcirc} \subseteq W^\star$ given as input, so as to compute the required estimates of the sender's expected utilities; and (ii) *exploitation* (Line 10), playing strategy $\mathbf{q}^\tau$ recommend by FULL-INFORMATION$(\cdot)$ for $I_\tau$.

Our main result is the proof that Algorithm 1 achieves the no-regret property. Formally:

**Theorem 4.** *Given an online Bayesian persuasion problem with partial feedback, there exist $W^\circ \subseteq W^\star$, $W^{\circledcirc} \subseteq W^\star$, and estimators $\tilde{u}^{\mathsf{s}}_{I_\tau}(\mathbf{w})$ such that Algorithm 1 provides the following regret bound:*

$$R^T \leq O\left(\frac{nm^{2/3}d\log^{1/3}(mn+d)}{T^{1/5}}\right).$$

In order to prove this result, we show that Algorithm 1 provides a regret bound that depends on the number $|W^{\circledcirc}|$ of signaling schemes used for exploration, the logarithm of $|W^\circ|$, and the range and bias of the estimators $\tilde{u}^{\mathsf{s}}_{I_\tau}(\mathbf{w})$. To do this, we extend a result shown by Balcan *et al.* [8, Lemma 6.2] to the more general case in which only *biased* utility estimators are available, rather than unbiased ones. This result can be generalized to any partial information setting (beyond online Bayesian persuasion).

In any block $I_\tau$ with $\tau \in [Z]$, for every $\mathbf{w} \in W^\circ$, we assume that Algorithm 1 has access to an estimator $\tilde{u}^{\mathsf{s}}_{I_\tau}(\mathbf{w})$ of the sender's average utility $u^{\mathsf{s}}_{I_\tau}(\mathbf{w}) = \frac{1}{|I_\tau|}\sum_{t\in[T]:t\in I} u^{\mathsf{s}}(\mathbf{w}, k^t)$ obtained by committing to $\mathbf{w}$ during the block $I_\tau$, with the following properties:

 (i) the *bias is bounded* by a given constant $\iota \in (0,1)$, *i.e.*, it holds $\left|u^{\mathsf{s}}_{I_\tau}(\mathbf{w}) - \mathbb{E}\left[\tilde{u}^{\mathsf{s}}_{I_\tau}(\mathbf{w})\right]\right| \leq \iota$;

 (ii) the *range is limited*, *i.e.*, there exists a $\eta \in \mathbb{R}$ such that $\tilde{u}^{\mathsf{s}}_{I_\tau}(\mathbf{w}) \in [-\eta, +\eta]$.

**Lemma 3.** *Suppose that Algorithm 1 has access to estimators $\tilde{u}^{\mathsf{s}}_{I_\tau}(\mathbf{w})$ with properties (i) and (ii) for some constants $\iota \in (0,1)$ and $\eta \in \mathbb{R}$, for every signaling scheme $\mathbf{w} \in W^\circ$ and block $I_\tau$ with $\tau \in [Z]$. Moreover, let $Z := T^{2/3}|W^{\circledcirc}|^{-2/3}\eta^{2/3}\log^{1/3}|W^\circ|$. Then, Algorithm 1 guarantees regret:*

$$R^T \leq O\left(\frac{|W^{\circledcirc}|^{1/3}\eta^{2/3}\log^{1/3}|W^\circ|}{T^{1/3}}\right) + O(\iota).$$

Lemma 3 shows that even if utility estimators have small bias, we can still hope for a no-regret algorithm. However, we have to guarantee that $W^{\circledcirc}$ has a polynomial size, and that the estimator has a limited range. These requirements can be achieved by estimating sender's utilities indirectly by means of other related estimates, at the cost of giving up on the unbiasedness of the estimators.

The key observation that allows to get the desired estimators $\tilde{u}^{\mathsf{s}}_{I_\tau}(\mathbf{w})$ by only exploring a polynomially-sized set $W^{\circledcirc}$ is that the utilities $u^{\mathsf{s}}_{I_\tau}(\mathbf{w})$ that we wish to estimate are *not* independent, but they all depend on the frequency of each receiver's type during block $I_\tau$. Thus, only these (polynomially many) quantities need to be estimated. In order to do so, we use the concept of *barycentric spanners* [4] (see Appendix D.2 for the details). A direct application of barycentric spanners to our setting would require being able to induce *any* receiver's posterior during the exploration phase. Unfortunately, this is not possible as the sender is forced to play consistent signaling schemes (see Equation (2)), which could prevent her from inducing certain posteriors. We achieve the goal of keeping the bias and the range of the estimators small by adopting the following two technical caveats:

 (i) we focus on posteriors that can be induced by a signaling scheme with at least some ('not too small') probability, which ensures that the resulting estimators have a limited range; and

 (ii) we restrict the full-information algorithm to signaling schemes $W^\circ \subseteq W^\star$ inducing a small number of posteriors, which guarantees to have estimators with a small bias.

We provide our complete technical results in Appendix D.

## 7 Discussion and future works

We proposed the online Bayesian persuasion framework as a natural extension of the original model by Kamenica and Gentzkow [23]. This is, to the best of our knowledge, the first work relaxing the

assumption that the sender has a perfect knowledge of the receiver's utility function. We proved that any no-regret algorithm for this setting has to require an exponential per-round running time, and we designed no-regret algorithms for the partial and full information feedback settings with adversarially chosen sequences of types. In the future, it would be interesting to study what happens if the receiver can play, at each round, an approximate best response ($\epsilon$-best response) to the sender signal. We conjecture that in this case it should be possible to build a no-regret algorithm with quasi-polynomial per-round running time.

## Broader Impact

Bayesian persuasion is a fascinating model that suffers from some limiting assumptions, which prevented a widespread use of the framework in practical applications. This work tries to amend one of such limitations, by relaxing the constraint that the sender has to have a perfect knowledge of the payoff structure of the game. This goes in the direction of developing a complete theory of *Bayesian persuasion from data* as a framework based solely on sender's and receiver's historical observations. In the future, an application of this framework at scale (*e.g.*, on large social platforms) could raise some societal challenges (see, *e.g.*, recent works on Bayesian persuasion as an election-manipulation tool). Therefore, future research in this direction should prioritize the study of how to protect receivers from excessive information garbling, and how to incentivize senders to work towards a socially-acceptable outcome.

## Acknowledgments and Disclosure of Funding

This work has been partially supported by the Italian MIUR PRIN 2017 Project ALGADIMAR "Algorithms, Games, and Digital Market".

## Footnotes

[3]int$(X)$ is the *interior* of set $X$ and $\Delta_X$ is the set of all probability distributions over $X$. Vectors are denoted by bold symbols. For any vector $\mathbf{x}$, the value of its $i$-th component is denoted by $x_i$.

[4]This assumption is customary in settings involving commitments, such as Stackelberg games [17, 18, 29].

[5]Throughout the paper, the set $\{1, \ldots, x\}$ is denoted by $[x]$.

[6]Theorem 2 can be obtained as a corollary of Theorem 6.2 by Roughgarden and Wang [31].

[7]$V(X)$ denotes the set of vertices of polytope $X$.

[8] The reduction is an extension of those proposed by Balcan *et al.* [8] and Awerbuch and Mansour [5].

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
