[Supplementary Material]

# A  Additional example

This example (see Figure 3) builds on the classical prosecutor/judge game by Kamenica and Gentzkow [23] described in Section 2.2. Here, the judge has two possible types. A judge of *type 1* gets payoff 1 for a just decision, and 0 otherwise. A judge of *type 2* has a worse perception of acquitting a guilty defendant, for which she gets $-1$. In this case, the computation of best-response regions is more involved because different judge's types yield different boundaries on the space of posteriors. Specifically, by Equation (4), $\hat{W}$ is the result of the intersection between the simplex $\Delta_{\hat{\Xi}}$ and the closed half-spaces specified by $[\boldsymbol{\xi}_1|\boldsymbol{\xi}_2|\boldsymbol{\xi}_3|\boldsymbol{\xi}_4] \cdot \mathbf{w} \geq \boldsymbol{\mu}$. The vertices of the resulting polytope are $\mathbf{w}_1 = (3/10, 0, 0, 7/10)^\top$, $\mathbf{w}_2 = (0, 9/10, 0, 1/10)^\top$, and $\mathbf{w}_3 = (0, 0, 3/5, 2/5)^\top$. Then, the new sender's action space can be restricted to $W^\star = \{\mathbf{w}_1, \mathbf{w}_2, \mathbf{w}_3\}$. [9]

Figure 3: ***Left***: A prosecutor/judge game with two types. When the judge is of type 2 she has a worse perception of acquitting a guilty defendant. ***Center***: A visual depiction of $\Xi_A^k$ and $\Xi_C^k$ for each possible type $k \in \{1, 2\}$. When $k = 2$, the judge is less inclined towards acquitting and, therefore, the best-response boundary is $\boldsymbol{\xi}_4$. When $k = 1$ (resp., $k = 2$) and the posterior is $\boldsymbol{\xi}_2$ (resp., $\boldsymbol{\xi}_4$), the judge is indifferent between acquitting and convicting the defendant. ***Right***: Best-response regions for the possible joint actions. When $\mathbf{a} = (C, A)$ we have $\Xi_\mathbf{a} = \varnothing$ because there is no posterior for which A is a best response for a receiver of type 1, and C is a best response for a receiver of type 2. We have $\hat{\Xi} = \{\boldsymbol{\xi}_1, \boldsymbol{\xi}_2, \boldsymbol{\xi}_3, \boldsymbol{\xi}_4\}$.

# B  Proofs omitted from Section 4

**Theorem 1.** *For every $0 \leq \alpha < 1$, it is* NP-*hard to compute an $\alpha$-optimal solution to* OPT-SIGNAL.

*Proof.* In order to prove Theorem 1, we resort to a result by Guruswami and Raghavendra [21] (see Theorem 5 below), which is about the following *promise problem* related to the satisfiability of a fraction of linear equations with rational coefficients and variables restricted to the hypercube.

**Definition 3** (LINEQ-MA$(1 - \zeta, \delta)$ by Guruswami and Raghavendra [21]). *For any two constants $\zeta, \delta \in \mathbb{R}$ satisfying $0 \leq \delta \leq 1 - \zeta \leq 1$,* LINEQ-MA$(1 - \zeta, \delta)$ *is the following promise problem: Given a set of linear equations $\mathbf{A}\mathbf{x} = \mathbf{c}$ over variables $\mathbf{x} \in \mathbb{Q}^{n_{\mathrm{var}}}$, with coefficients $\mathbf{A} \in \mathbb{Q}^{n_{\mathrm{eq}} \times n_{\mathrm{var}}}$ and $\mathbf{c} \in \mathbb{Q}^{n_{\mathrm{var}}}$, distinguish between the following two cases:*

- *there exists a vector $\hat{\mathbf{x}} \in \{0, 1\}^{n_{\mathrm{var}}}$ that satisfies at least a fraction $1 - \zeta$ of the equations;*

- *every possible vector $\mathbf{x} \in \mathbb{Q}^{n_{\mathrm{var}}}$ satisfies less than a fraction $\delta$ of the equations.*

**Theorem 5** (Guruswami and Raghavendra [21]). *For all valid $\zeta, \delta > 0$,* LINEQ-MA$(1 - \zeta, \delta)$ *is* NP-*hard.*

We introduce a reduction from LINEQ-MA$(1 - \zeta, \delta)$ to OPT-SIGNAL, showing the following:

- *Completeness*: If an instance of LINEQ-MA$(1 - \zeta, \delta)$ admits a $1 - \zeta$ fraction of satisfiable equations when variables are restricted to lie the hypercube $\{0, 1\}^{n_{\mathrm{var}}}$, then an optimal solution to OPT-SIGNAL provides the sender with an expected utility at least of $1 - 2\zeta$;

- *Soundness*: If at most a $\delta$ fraction of the equations can be satisfied, then an optimal solution to OPT-SIGNAL has sender's expected utility at most $\delta$.

Since $\zeta$ and $\delta$ can be arbitrary (with $0 \le \delta \le 1 - \zeta \le 1$), the two properties above immediately prove the result. In the rest of the proof, given a vector of variables $\mathbf{x} \in \mathbb{Q}^{n_{\text{var}}}$, for $i \in [n_{\text{var}}]$, we denote with $x_i$ the component corresponding to the $i$-th variable. Similarly, for $j \in [n_{\text{eq}}]$, $c_j$ is the $j$-th component of the vector $\mathbf{c}$, whereas, for $i \in [n_{\text{var}}]$ and $j \in [n_{\text{eq}}]$, the $(j,i)$-entry of $\mathbf{A}$ is denoted by $A_{ji}$.

**Reduction**  As a preliminary step, we normalize the coefficients by letting $\bar{\mathbf{A}} := \frac{1}{\tau}\mathbf{A}$ and $\bar{\mathbf{c}} := \frac{1}{\tau}\mathbf{c}$, where we let $\tau := 2\max\left\{\max_{i\in[n_{\text{var}}],j\in[n_{\text{eq}}]} A_{ji}, \max_{j\in[n_{\text{eq}}]} c_j, n_{\text{var}}^2\right\}$. It is easy to see that the normalization preserves the number of satisfiable equations. Formally, the number of satisfied equations of $\mathbf{A}\mathbf{x} = \mathbf{c}$ is equal to the number of satisfied equations of $\bar{\mathbf{A}}\bar{\mathbf{x}} = \bar{\mathbf{c}}$, where $\bar{\mathbf{x}} = \frac{1}{\tau}\mathbf{x}$. For every variable $i \in [n_{\text{var}}]$, we define a state of nature $\theta_i \in \Theta$. Moreover, we introduce an additional state $\theta_0 \in \Theta$. The prior distribution $\mu \in \text{int}(\Delta_\Theta)$ is defined in such a way that $\mu_{\theta_i} = \frac{1}{n_{\text{var}}^2}$ for every $i \in [n_{\text{var}}]$, while $\mu_{\theta_0} = \frac{n_{\text{var}}-1}{n_{\text{var}}}$ (notice that $\sum_{\theta\in\Theta}\mu_\theta = 1$). We define a receiver's type $k_j \in \mathcal{K}$ for each equation $j \in [n_{\text{eq}}]$ (recall that the distribution over receiver's types $\rho \in \Delta_\mathcal{K}$ is uniform by definition of OPT-SIGNAL). The receiver has three actions available, namely $\mathcal{A} := \{a_0, a_1, a_2\}$, whereas, for every $k_j \in \mathcal{K}$, the utilities of type $k_j$ are $u_{\theta_i}^{k_j}(a_0) = \frac{1}{2}$, $u_{\theta_i}^{k_j}(a_1) = \frac{1}{2} - \bar{A}_{ji} + \bar{c}_j$, and $u_{\theta_i}^{k_j}(a_2) = \frac{1}{2} + \bar{A}_{ji} - \bar{c}_j$ for every $i \in [n_{\text{var}}]$, while $u_{\theta_0}^{k_j}(a_0) = \frac{1}{2}$, $u_{\theta_0}^{k_j}(a_1) = \frac{1}{2} + \bar{c}_j$, and $u_{\theta_0}^{k_j}(a_2) = \frac{1}{2} - \bar{c}_j$. Finally, the sender's utility is $1$ when the receiver plays $a_0$, while it is $0$ otherwise, independently of the state of nature. Formally, $u_\theta^s(a_0) = 1$ and $u_\theta^s(a_1) = u_\theta^s(a_2) = 0$ for every $\theta \in \Theta$.

**Completeness**  Suppose there exists a vector $\hat{\mathbf{x}} \in \{0,1\}^{n_{\text{var}}}$ such that at least a fraction $1 - \zeta$ of the equations in $\mathbf{A}\hat{\mathbf{x}} = \mathbf{c}$ are satisfied. Let $X^1 \subseteq [n_{\text{var}}]$ be the set of variables $i \in [n_{\text{var}}]$ with $x_i = 1$, while $X^0 := [n_{\text{var}}] \setminus X^1$. Given the definition of $\bar{\mathbf{A}}$ and $\bar{\mathbf{c}}$, there exists a vector $\bar{\mathbf{x}} \in \{0, \frac{1}{\tau}\}^{n_{\text{var}}}$ such that at least a fraction $1 - \zeta$ of the equations in $\bar{\mathbf{A}}\bar{\mathbf{x}} = \bar{\mathbf{c}}$ are satisfied, and, additionally, $\bar{x}_i = \frac{1}{\tau}$ for all the variables in $i \in X^1$, while $\bar{x}_i = 0$ whenever $i \in X^0$. Let us consider an (indirect) signaling scheme $\phi : \Theta \to \Delta_\mathcal{S}$ defined for the set of signals $\mathcal{S} := \{s_1, s_2\}$. Let $q := \frac{n_{\text{var}}(n_{\text{var}}-1)}{\tau - |X^1|}$. For every $i \in [n_{\text{var}}]$, we define $\phi_{\theta_i}(s_1) = q$ and $\phi_{\theta_i}(s_2) = 1 - q$ if $i \in X^1$, while $\phi_{\theta_i}(s_1) = 0$ and $\phi_{\theta_i}(s_2) = 1$ otherwise. Moreover, we let $\phi_{\theta_0}(s_1) = 1$ and $\phi_{\theta_0}(s_2) = 0$. Now, let us take the receiver's posterior $\boldsymbol{\xi}^1 \in \Delta_\Theta$ induced by signal $s_1$. Let $h := \frac{\frac{q}{n_{\text{var}}^2}}{\sum_{i\in X^1}\frac{q}{n_{\text{var}}^2} + \frac{n_{\text{var}}-1}{n_{\text{var}}}}$. Then, using the definition of $\boldsymbol{\xi}^1$, it is easy to check that $\xi_{\theta_i}^1 = h$ for every $i \in X^1$, $\xi_{\theta_i}^1 = 0$ for every $i \in X^0$, while $\xi_{\theta_0}^1 = \frac{\frac{n_{\text{var}}-1}{n_{\text{var}}}}{\sum_{i\in X^1}\frac{q}{n_{\text{var}}^2} + \frac{n_{\text{var}}-1}{n_{\text{var}}}} = 1 - h|X^1|$. Next, we prove that given the posterior $\boldsymbol{\xi}^1$ at least a fraction $1 - \zeta$ of the receiver's types has action $a_0$ as a best response, implying that the expected utility of the sender is equal to $\frac{1}{n}\sum_{k\in\mathcal{K}} u^s(\phi, k) \ge \frac{n-1}{n}(1-\zeta) \ge 1 - 2\zeta$, which holds for $n$ large enough. For each satisfied equality $j \in [n_{\text{eq}}]$ in $\bar{\mathbf{A}}\bar{\mathbf{x}} = \bar{\mathbf{c}}$, the receiver of type $k_j \in \mathcal{K}$ experiences a utility of $\sum_{\theta\in\Theta}\xi_\theta^1 u_\theta^{k_j}(a_0) = \frac{1}{2}$ by playing action $a_0$. Instead, the utility she gets by playing $a_1$ is defined as follows:

$$\sum_{\theta\in\Theta}\xi_\theta^1 u_\theta^{k_j}(a_1) = \sum_{i\in X^1} h\left(\frac{1}{2} - \bar{A}_{ji} + \bar{c}_j\right) + \xi_{\theta_0}^1\left(\frac{1}{2} + \bar{c}_j\right) =$$

$$= h|X^1|\left(\frac{1}{2} + \bar{c}_j\right) - h\sum_{i\in X^1}\bar{A}_{ji} + \left(1 - h|X^1|\right)\left(\frac{1}{2} + \bar{c}_j\right) =$$

$$= \frac{1}{2} + \bar{c}_j - h\sum_{i\in X^1}\bar{A}_{ji} = \frac{1}{2} + \bar{c}_j - \frac{1}{\tau}\sum_{i\in X^1}\bar{A}_{ji} = \frac{1}{2},$$

where the second to last equality holds since $h = \frac{1}{\tau}$ (by definition of $h$ and $q$), while the last equality follows from the fact that the $j$-th equation is satisfied, and, thus, $\frac{1}{\tau}\sum_{i\in X^1}\bar{A}_{ji} = \bar{c}_j$ (recall that $\bar{x}_i = \frac{1}{\tau}$ for all $i \in X^1$). Using similar arguments, we can write $\sum_{\theta\in\Theta}\xi_\theta^1 u_\theta^{k_j}(a_2) = \frac{1}{2}$, which concludes the completeness proof.

**Soundness**  Suppose, by contradiction, that there exists a signaling scheme $\phi : \Theta \to \Delta_\mathcal{S}$ providing the sender with an expected utility greater than $\delta$. This implies, by an averaging argument, that there exists a signal inducing a posterior $\boldsymbol{\xi} \in \Delta_\Theta$ in which at least a fraction $\delta$ of the receiver's types best responds by playing action $a_0$. Let $\mathcal{K}^1 \subseteq \mathcal{K}$ be the set of such reviver's types. For every receiver's type $k_j \in \mathcal{K}$, it holds $\sum_{\theta\in\Theta}\xi_\theta u_\theta^{k_j}(a_0) = \frac{1}{2}$.

Moreover, it is the case that:

$$\sum_{\theta \in \Theta} \xi_\theta u_\theta^{k_j}(a_1) = \sum_{i \in [n_{\mathrm{var}}]} \xi_{\theta_i} \left( \frac{1}{2} - \bar{A}_{ji} + \bar{c}_j \right) + \xi_{\theta_0} \left( \frac{1}{2} + \bar{c}_j \right) = \frac{1}{2} + \bar{c}_j - \sum_{i \in [n_{\mathrm{var}}]} \xi_{\theta_i} \bar{A}_{ji}.$$

Similarly, it holds:

$$\sum_{\theta \in \Theta} \xi_\theta u_\theta^{k_j}(a_2) = \frac{1}{2} - \bar{c}_j + \sum_{i \in [n_{\mathrm{var}}]} \xi_{\theta_i} \bar{A}_{ji}.$$

By assumption, for every type $k_j \in \mathcal{K}^1$, it is the case that $\sum_{\theta \in \Theta} \xi_\theta u_\theta^{k_j}(a_0) \geq \sum_{\theta \in \Theta} \xi_\theta u_\theta^{k_j}(a_1)$, which implies that $\bar{c}_j - \sum_{i \in [n_{\mathrm{var}}]} \xi_{\theta_i} \bar{A}_{ji} \leq 0$, whereas $\sum_{\theta \in \Theta} \xi_\theta u_\theta^{k_j}(a_0) \geq \sum_{\theta \in \Theta} \xi_\theta u_\theta^{k_j}(a_2)$, implying $-\bar{c}_j + \sum_{i \in [n_{\mathrm{var}}]} \xi_{\theta_i} \bar{A}_{ji} \leq 0$. Thus, $\sum_{i \in [n_{\mathrm{var}}]} \xi_{\theta_i} \bar{A}_{ji} = \bar{c}_j$ for every $j \in [n_{\mathrm{eq}}]$ such that $k_j \in \mathcal{K}^1$ and the vector $\hat{\mathbf{x}} \in \mathbb{Q}^{n_{\mathrm{var}}}$ with $\hat{x}_i = \xi_{\theta_i}$ for all $i \in [n_{\mathrm{var}}]$ satisfies at least a fraction $\delta$ of the equations, reaching a contradiction. $\qquad\square$

## C  Proofs omitted from Section 5

**Lemma 1.** *For every sequence of receiver's types $\boldsymbol{k} = \{k^t\}_{t \in [T]}$, it holds*

$$\max_{\mathbf{w} \in W} \sum_{t=1}^{T} u^{\mathsf{s}}(\mathbf{w}, k^t) = \max_{\mathbf{w}^\star \in W^\star} \sum_{t=1}^{T} u^{\mathsf{s}}(\mathbf{w}^\star, k^t).$$

*Proof.* The idea to prove the lemma is the following: any posterior distribution $\boldsymbol{\xi}$ in $\mathrm{supp}(\mathbf{w})$ can be represented as the convex combination of elements of $\hat{\Xi}$. We denote such convex combination by $\mathbf{w}^{\boldsymbol{\xi}} \in \Delta_{\hat{\Xi}}$. We define a new signaling scheme $\mathbf{w}^\star \in \Delta_{\hat{\Xi}}$ as follows:

$$w_{\boldsymbol{\xi}'}^\star := \sum_{\substack{\boldsymbol{\xi} \in \mathrm{supp}(\mathbf{w}): \\ \boldsymbol{\xi}' \in \mathrm{supp}(\mathbf{w}^{\boldsymbol{\xi}})}} w_{\boldsymbol{\xi}} w_{\boldsymbol{\xi}'}^{\boldsymbol{\xi}}, \quad \text{for each} \quad \boldsymbol{\xi}' \in \hat{\Xi}. \tag{6}$$

Since $\mathbf{w}$ is consistent (*i.e.,* $\mathbf{w} \in W$) we have by construction that $\mathbf{w}^\star$ is consistent, and therefore $\mathbf{w}^\star \in \hat{W}$. Finally, we show that $\mathbf{w}^\star$ guarantees to the sender an expected utility which is greater than or equal to that achieved via $\mathbf{w}$. The crucial point here is showing that whenever the decomposition over $\hat{\Xi}$ involves a vertex (*i.e.,* a posterior) where the receiver is indifferent between two or more actions, her/his choice does not damage the sender. This happens at the boundaries of best-response regions (see, *e.g.,* what happens at $\boldsymbol{\xi}_2$ and $\boldsymbol{\xi}_4$ in the example of Figure 3). The sender's expected utility is a linear function of the signaling scheme $\mathbf{w}^\star$. Therefore, the sender can limit her attention to $W^\star$, since her/his maximum expected utility is attained at one of the vertices of $\hat{W}$.

Consider a posterior $\boldsymbol{\xi} \in \Xi$ and let $\mathbf{a} = \{b_{\boldsymbol{\xi}}^k\}_{k \in \mathcal{K}}$ (*i.e.,* $\mathbf{a}$ is the tuple specifying the best-response action under posterior $\boldsymbol{\xi}$ for each receiver's type $k$). Tuple $\mathbf{a}$ defines polytope $\Xi_{\mathbf{a}} \subseteq \Xi$. By Carathéodory's theorem, any $\boldsymbol{\xi} \in \Xi_{\mathbf{a}}$ is the convex combination of a finite number of points in $\Xi_{\mathbf{a}}$. Specifically, there exists $\mathbf{w}^{\boldsymbol{\xi}} \in \Delta_{V(\Xi_{\mathbf{a}})}$ such that, for each $\theta \in \Theta$, $\sum_{\boldsymbol{\xi}' \in V(\Xi_{\mathbf{a}})} w_{\boldsymbol{\xi}'}^{\boldsymbol{\xi}} \xi_\theta' = \xi_\theta$.

Let $\mathbf{w} \in \hat{W}$ (*i.e.,* $\mathbf{w}$ is consistent). By following Equation (6), we define a distribution $\mathbf{w}^\star$ such that, for each $\boldsymbol{\xi}' \in \hat{\Xi}$,

$$w_{\boldsymbol{\xi}'}^\star := \sum_{\substack{\boldsymbol{\xi} \in \mathrm{supp}(\mathbf{w}): \\ \boldsymbol{\xi}' \in \mathrm{supp}(\mathbf{w}^{\boldsymbol{\xi}})}} w_{\boldsymbol{\xi}} w_{\boldsymbol{\xi}'}^{\boldsymbol{\xi}}.$$

By construction, $\mathbf{w}^\star$ is a well-defined convex combination of elements of $\hat{\Xi}$. Moreover, since $\mathbf{w}$ is consistent, the same holds true for $\mathbf{w}^\star$, which implies $\mathbf{w}^\star \in \hat{W}$.

Fix a type $k \in \mathcal{K}$ and a posterior $\boldsymbol{\xi} \in \Xi$, and let $\mathbf{a}$ be defined as the tuple specifying the best response under $\boldsymbol{\xi}$ for each $k$. At each posterior $\boldsymbol{\xi}' \in V(\Xi_{\mathbf{a}})$, the receiver plays $b_{\boldsymbol{\xi}'}^k$. The following holds:

$$b_{\boldsymbol{\xi}'}^k \in \arg\max_{a' \in \mathcal{B}_{\boldsymbol{\xi}'}^k} \sum_{\theta \in \Theta} \xi_\theta' u_\theta^{\mathsf{s}}(a') \geq \sum_{\theta \in \Theta} \xi_\theta' u_\theta^{\mathsf{s}}(b_{\boldsymbol{\xi}}^k), \tag{7}$$

where the inequality holds because, by construction, $b_{\boldsymbol{\xi}}^k \in \mathcal{B}_{\boldsymbol{\xi}'}^k$. Therefore, we can show that the sender's expected utility when decomposing $\boldsymbol{\xi}$ as $\mathbf{w}^{\boldsymbol{\xi}} \in \Delta_{V(\Xi_\mathbf{a})}$ is guaranteed to be greater than or equal to the expected utility under $\boldsymbol{\xi}$. Specifically,

$$
\begin{aligned}
\sum_{\boldsymbol{\xi}' \in V(\Xi_\mathbf{a})} w_{\boldsymbol{\xi}'}^{\boldsymbol{\xi}} u^{\mathsf{s}}(\boldsymbol{\xi}', k) &= \sum_{\boldsymbol{\xi}' \in V(\Xi_\mathbf{a})} w_{\boldsymbol{\xi}'}^{\boldsymbol{\xi}} \sum_{\theta \in \Theta} \xi_\theta' u_\theta^{\mathsf{s}}(b_{\boldsymbol{\xi}'}^k) \\
&\geq \sum_{\boldsymbol{\xi}' \in V(\Xi_\mathbf{a})} w_{\boldsymbol{\xi}'}^{\boldsymbol{\xi}} \sum_{\theta \in \Theta} \xi_\theta' u_\theta^{\mathsf{s}}(b_{\boldsymbol{\xi}}^k) && \text{(By Equation (7))} \\
&= \sum_{\theta \in \Theta} \xi_\theta u_\theta^{\mathsf{s}}(b_{\boldsymbol{\xi}}^k) && \text{(By definition of } \mathbf{w}^{\boldsymbol{\xi}}) \\
&= u^{\mathsf{s}}(\boldsymbol{\xi}, k).
\end{aligned}
$$

Let $\mathbf{w} \in W$ be the best-in-hindsight signaling scheme. We show that, for any sequence of receiver's types $\mathbf{k} = \{k^t\}_{t \in [T]}$, the sender's expected utility achieved via $\mathbf{w}$ is matched by the expected utility guaranteed by $\mathbf{w}^\star \in \hat{W}$ defined as in Equation (6). We have

$$
\begin{aligned}
\sum_{t \in [T]} \sum_{\boldsymbol{\xi} \in \text{supp}(\mathbf{w}^\star)} w_{\boldsymbol{\xi}}^\star u^{\mathsf{s}}(\boldsymbol{\xi}, k^t) &= \sum_{t \in [T]} \sum_{\boldsymbol{\xi} \in \text{supp}(\mathbf{w}^\star)} \sum_{\substack{\boldsymbol{\xi}' \in \text{supp}(\mathbf{w}): \\ \boldsymbol{\xi} \in \text{supp}(\mathbf{w}^{\boldsymbol{\xi}'})}} w_{\boldsymbol{\xi}'} w_{\boldsymbol{\xi}}^{\boldsymbol{\xi}'} u^{\mathsf{s}}(\boldsymbol{\xi}, k^t) \\
&= \sum_{t \in [T]} \sum_{\boldsymbol{\xi}' \in \text{supp}(\mathbf{w})} w_{\boldsymbol{\xi}'} \sum_{\boldsymbol{\xi} \in \text{supp}(\mathbf{w}^{\boldsymbol{\xi}'})} w_{\boldsymbol{\xi}}^{\boldsymbol{\xi}'} u^{\mathsf{s}}(\boldsymbol{\xi}, k^t) \\
&\geq \sum_{t \in [T]} \sum_{\boldsymbol{\xi}' \in \text{supp}(\mathbf{w})} u^{\mathsf{s}}(\boldsymbol{\xi}', k^t) \\
&= \sum_{t \in [T]} u^{\mathsf{s}}(\mathbf{w}, k^t).
\end{aligned}
$$

Finally, since $\sum_{t \in [T]} u^{\mathsf{s}}(\mathbf{w}^\star, k^t) = \sum_{t \in [T]} \sum_{\boldsymbol{\xi} \in \text{supp}(\mathbf{w}^\star)} w_{\boldsymbol{\xi}}^\star u^{\mathsf{s}}(\boldsymbol{\xi}, k^t)$ is a linear function in the signaling scheme $\mathbf{w}^\star$, its maximum is attained at a vertex of $\hat{W}$. This concludes the proof. $\qquad \square$

**Lemma 2.** *The size of $W^\star$ is $|W^\star| \in O\left((n\,m^2 + d)^d\right)$.*

*Proof.* By definition, for any $\mathbf{a} = (a^k)_{k \in \mathcal{K}}$, $W_\mathbf{a} \subseteq \Xi$. Then, each $\mathbf{w} \in V(W_\mathbf{a})$ is an extreme point of a $(d-1)$-dimensional convex polytope, and therefore the point lies at the intersection of $(d-1)$ linearly independent defining half-spaces of the polytope. Now, to provide a bound for $|\hat{\Xi}|$ we first compute the number of half-spaces separating best-response regions corresponding to different actions. For each type $k \in \mathcal{K}$, there are at most $\binom{m}{2}$ half-spaces each separating $W_a^k$ and $W_{a'}^k$ for two actions $a \neq a'$. Then, in order to take all the incentive constraints into account, we have to sum over all possible reveiver's types, obtaining $O(n\,m^2)$ half-spaces. The set $\hat{\Xi}$ is the result of the intersection between the region defined by such half-spaces, and the $d$ constraints defining the simplex. Each extreme point of the polytope defined by points in $\hat{\Xi}$ lies at the intersection of $d-1$ half-spaces. Therefore, there are at most $\binom{nm^2 + d}{d-1} \in O\left((nm^2 + d)^d\right)$ such extreme points. The convex polytope $\hat{W}$ is the result of the intersection between the simplex defined over $\hat{\Xi}$, which has $O\left((nm^2 + d)^d\right)$ extreme points, and $d$ half-spaces defining consistency constraints. Then, $\hat{W}$ has a number of extreme points which is less than or equal to $O\left((nm^2 + d)^d\right)$. $\qquad \square$

**Theorem 3.** *Given an online Bayesian persuasion problem with full information feedback, there exists an online algorithm such that, for every sequence of receiver's types $\boldsymbol{k} = \{k^t\}_{t \in [T]}$:*

$$
R^T \leq O\left(\sqrt{\frac{d \log(n\,m^2 + d)}{T}}\right).
$$

*Proof.* We employ an arbitrary algorithm satisfying $R^T \leq O\left(\eta\sqrt{\log |A|/T}\right)$ with action set $A = W^\star$. Let $\mathbf{w}^* \in W$ be the sender-optimal signaling scheme in hindsight. Then,

$$\sum_{t\in[T]} \mathbb{E}[u^s(\mathbf{w}^t, k^t)] \geq \sum_{t\in[T]} u^s(\mathbf{w}^*, k^t) - O\left(\sqrt{T \log |W^\star|}\right)$$

$$\geq \sum_{t\in[T]} u^s(\mathbf{w}^*, k^t) - O\left(\sqrt{T \log (nm^2 + d)^d}\right) \qquad \text{(By Lemma 2)}$$

$$= \sum_{t\in[T]} u^s(\mathbf{w}^*, k^t) - O\left(\sqrt{Td \log (nm^2 + d)}\right).$$

This completes the proof. $\qquad\square$

## D  Additional results on the partial information feedback setting

Appendix D.1 reports the proof of Lemma 3, which shows a regret bound for the reduction from partial information to full information that exploits biased estimators. Appendix D.2 provides a detailed treatment on how Algorithm 1 computes the required sender's utility estimates. Finally, Appendix D.3 concludes with the proof of Theorem 4.

### D.1  Proof of Lemma 3

**Lemma 3.** *Suppose that Algorithm 1 has access to estimators $\tilde{u}^s_{I_\tau}(\mathbf{w})$ with properties (i) and (ii) for some constants $\iota \in (0,1)$ and $\eta \in \mathbb{R}$, for every signaling scheme $\mathbf{w} \in W^\circ$ and block $I_\tau$ with $\tau \in [Z]$. Moreover, let $Z := T^{2/3} |W^\circledcirc|^{-2/3} \eta^{2/3} \log^{1/3} |W^\circ|$. Then, Algorithm 1 guarantees regret:*

$$R^T \leq O\left(\frac{|W^\circledcirc|^{1/3} \eta^{2/3} \log^{1/3} |W^\circ|}{T^{1/3}}\right) + O\left(\iota\right).$$

*Proof.* In order to prove the desired regret bound for Algorithm 1, we rely on two crucial observations:

- during the exploration phase of each block $I_\tau$ with $\tau \in [Z]$, *i.e.*, the iterations $t_1, \ldots, t_{|W^\circledcirc|}$, the algorithm plays a strategy $\mathbf{q}^t \neq \mathbf{q}^\tau$, where $\mathbf{q}^\tau$ is the last strategy recommended by FULL-INFORMATION($\cdot$), resulting in a corresponding utility loss that can be as large as $-1$ (since the utilities are in the range $[0, 1]$);

- running the full-information no-regret algorithm (*i.e.*, the sub-procedure FULL-INFORMATION($\cdot$)) using biased estimates of the sender's utilities (rather than their real values) results in the regret bound being worsened by only a term that is proportional to the bias $\iota$ of the adopted estimators.

In the following, we denote with $R^Z_{\text{full}}$ the cumulative regret achieved by FULL-INFORMATION($\cdot$), where we remark the fact that each block $I_\tau$ simulates a single iteration of the full information setting, and, thus, the number of iterations for the full-information algorithm is $Z$ rather than $T$. Formally, we have the following definition:

$$R^Z_{\text{full}} := \max_{\mathbf{w}\in W^\circ} \sum_{\tau\in[Z]} \tilde{u}^s_{I_\tau}(\mathbf{w}) - \sum_{\tau\in[Z]} \sum_{\mathbf{w}\in W^\circ} q^\tau_{\mathbf{w}} \tilde{u}^s_{I_\tau}(\mathbf{w}),$$

where we notice that the regret is computed with respect to the estimates $\tilde{u}^s_{I_\tau}(\mathbf{w})$ of the sender's average utilities $u^s_{I_\tau}(\mathbf{w})$ experienced in each block $I_\tau$, defined as $u^s_{I_\tau}(\mathbf{w}) = \frac{1}{|I_\tau|}\sum_{t\in I_\tau} u^s(\mathbf{w}, k^t)$ for every $\mathbf{w} \in W^\circ$. We also remark that the full-information algorithm is run on a subset $W^\circ \subseteq W^\star$ of signaling schemes, and, thus, the regret $R^Z_{\text{full}}$ is defined with respect to them. Moreover, from Section 5, we know that there exists an algorithm satisfying the regret bound $R^Z_{\text{full}} \leq O\left(\eta\sqrt{Z \log |W^\circ|}\right)$, where $\eta$ is the range of the utility values observed by the algorithm that, in our case, corresponds to the range of the estimates observed by the algorithm, which is limited thanks to property (ii) of the estimators.

In order to prove the result, we also need the following relation, which holds for every $\tau \in [Z]$ and signaling scheme $\mathbf{w} \in W^\circ$:

$$\sum_{t\in I_\tau} u^s(\mathbf{w}, k^t) = |I_\tau| u^s_{I_\tau}(\mathbf{w}) \geq |I_\tau|\left(\mathbb{E}[\tilde{u}^s_{I_\tau}] - \iota\right) = \frac{T}{Z}\left(\mathbb{E}[\tilde{u}^s_{I_\tau}] - \iota\right), \qquad (8)$$

where the first equality holds by definition, the inequality holds thanks to property (i) of the estimators, while the last equality is given by $|I_\tau| = \frac{T}{Z}$.

Letting $U$ be the sender's expected utility achieved by playing according to Algorithm 1, the following relations hold:

$$\frac{1}{T}U := \frac{1}{T} \sum_{\tau \in [Z]} \sum_{t \in I_\tau} \sum_{\mathbf{w} \in W^\circ} q_{\mathbf{w}}^t u^{\mathsf{s}}(\mathbf{w}, k^t)$$

$$\geq \frac{1}{T} \sum_{\tau \in [Z]} \sum_{\mathbf{w} \in W^\circ} q_{\mathbf{w}}^\tau \sum_{t \in I_\tau} u^{\mathsf{s}}(\mathbf{w}, k^t) - \frac{|W^\odot|Z}{T} \qquad (\mathbf{q}^t \neq \mathbf{q}^\tau \text{ in } |W^\odot| \text{ iterations and max. loss} = -1)$$

$$\geq \frac{1}{T} \sum_{\tau \in [Z]} \sum_{\mathbf{w} \in W^\circ} q_{\mathbf{w}}^\tau \frac{T}{Z} \left( \mathbb{E}\left[\tilde{u}_{I_\tau}^{\mathsf{s}}(\mathbf{w})\right] - \iota \right) - \frac{|W^\odot|Z}{T} \qquad \text{(By Equation (8))}$$

$$= \frac{1}{Z} \sum_{\tau \in [Z]} \sum_{\mathbf{w} \in W^\circ} q_{\mathbf{w}}^\tau \left( \mathbb{E}\left[\tilde{u}_{I_\tau}^{\mathsf{s}}(\mathbf{w})\right] - \iota \right) - \frac{|W^\odot|Z}{T}$$

$$= \frac{1}{Z} \sum_{\tau \in [Z]} \sum_{\mathbf{w} \in W^\circ} q_{\mathbf{w}}^\tau \mathbb{E}\left[\tilde{u}_{I_\tau}^{\mathsf{s}}(\mathbf{w})\right] - \iota - \frac{|W^\odot|Z}{T} \qquad \left(\text{Since } \sum_{\tau \in [Z]} \sum_{\mathbf{w} \in W^\circ} q_{\mathbf{w}}^\tau = Z, \text{ being } \mathbf{q}^\tau \in \Delta_{W^\circ}\right)$$

$$= \frac{1}{Z} \mathbb{E}\left[ \sum_{\tau \in [Z]} \sum_{\mathbf{w} \in W^\circ} q_{\mathbf{w}}^\tau \tilde{u}_{I_\tau}^{\mathsf{s}}(\mathbf{w}) \right] - \iota - \frac{|W^\odot|Z}{T}$$

$$= \frac{1}{Z} \mathbb{E}\left[ \max_{\mathbf{w} \in W^\circ} \sum_{\tau \in Z} \tilde{u}_{I_\tau}^{\mathsf{s}}(\mathbf{w}) - R_{\text{full}}^Z \right] - \iota - \frac{|W^\odot|Z}{T} \qquad (\text{Definition of } R_{\text{full}}^Z)$$

$$\geq \frac{1}{Z} \max_{\mathbf{w} \in W^\circ} \sum_{\tau \in [Z]} \mathbb{E}\left[\tilde{u}_{I_\tau}^{\mathsf{s}}(\mathbf{w})\right] - \frac{1}{Z} R_{\text{full}}^Z - \iota - \frac{|W^\odot|Z}{T} \qquad (\text{Jensen's inequality})$$

$$\geq \frac{1}{Z} \max_{\mathbf{w} \in W^\circ} \sum_{\tau \in [Z]} \left( u_{I_\tau}^{\mathsf{s}}(\mathbf{w}) - \iota \right) - \frac{1}{Z} R_{\text{full}}^Z - \iota - \frac{|W^\odot|Z}{T} \qquad (\text{By property (i)})$$

$$= \frac{1}{Z} \max_{\mathbf{w} \in W^\circ} \sum_{\tau \in [Z]} u_{I_\tau}^{\mathsf{s}}(\mathbf{w}) - \iota - \frac{1}{Z} R_{\text{full}}^Z - \iota - \frac{|W^\odot|Z}{T}$$

$$= \frac{1}{Z} \max_{\mathbf{w} \in W^\circ} \frac{Z}{T} \sum_{\tau \in [Z]} \sum_{t \in I_\tau} u^{\mathsf{s}}(\mathbf{w}, k^t) - \frac{1}{Z} R_{\text{full}}^Z - 2\iota - \frac{|W^\odot|Z}{T} \qquad (\text{By def. of } u_{I_\tau}^{\mathsf{s}}(\mathbf{w}) \text{ and } |I_\tau| = \frac{T}{Z})$$

$$= \frac{1}{T} \max_{\mathbf{w} \in W^\circ} \sum_{\tau \in [Z]} \sum_{t \in I_\tau} u^{\mathsf{s}}(\mathbf{w}, k^t) - \frac{1}{Z} R_{\text{full}}^Z - 2\iota - \frac{|W^\odot|Z}{T}$$

$$= \frac{1}{T} \max_{\mathbf{w} \in W^\circ} \sum_{t \in [T]} u^{\mathsf{s}}(\mathbf{w}, k^t) - \frac{1}{Z} R_{\text{full}}^Z - 2\iota - \frac{|W^\odot|Z}{T} =$$

$$\geq \frac{1}{T} \max_{\mathbf{w} \in W^\circ} \sum_{t \in [T]} u^{\mathsf{s}}(\mathbf{w}, k^t) - \frac{1}{Z} O\left( \eta \sqrt{Z \log |W^\circ|} \right) - 2\iota - \frac{|W^\odot|Z}{T}$$

$$\geq \frac{1}{T} \max_{\mathbf{w} \in W^\circ} \sum_{t \in [T]} u^{\mathsf{s}}(\mathbf{w}, k^t) - O\left( \frac{|W^\odot|^{1/3} \eta^{2/3} \log^{1/3} |W^\circ|}{T^{1/3}} \right) - 2\iota - \frac{|W^\odot|^{1/3} \eta^{2/3} \log^{1/3} |W^\circ|}{T^{1/3}}$$

$$\geq \frac{1}{T} \max_{\mathbf{w} \in W^\circ} \sum_{t \in [T]} u^{\mathsf{s}}(\mathbf{w}, k^t) - O\left( \frac{|W^\odot|^{1/3} \eta^{2/3} \log^{1/3} |W^\circ|}{T^{1/3}} \right) - O(\iota)$$

By using the definition of the regret $R^T$ of Algorithm 1, we get the statement. $\qquad \square$

## D.2 Details on sender's average utilities estimation

In the following, we show in details how to compute the estimates needed by Algorithm 1 by using random samples from a polynomially-sized set $W^{\circledcirc} \subseteq W^{\star}$. Let us recall that, during each block $I_\tau$ with $\tau \in [Z]$, Algorithm 1 needs to compute the estimators $\tilde{u}_{I_\tau}^{\mathsf{s}}(\mathbf{w})$ of $u_{I_\tau}^{\mathsf{s}}(\mathbf{w}) = \frac{1}{|I_\tau|}\sum_{t \in I_\tau} u^{\mathsf{s}}(\mathbf{w}, k^t)$ for all the signaling schemes $\mathbf{w} \in W^{\circ}$ (Line 13). Notice that the set $W^{\circ} \subseteq W^{\star}$ is defined (as shown in Lemma 6) in order to be able to build estimators with the desired properties (i) and (ii).

As discussed in Section 6, the key insight that allows us to get the required estimates by using only a polynomial number of random samples is that the utilities to be estimated are *not* independent. This is because they depend on the frequencies of the receiver's actions during bock $I_\tau$, which depend, in turn, on the frequencies of the receiver's types. Thus, the goal is to devise estimators for the frequencies of the receiver's types during each block $I_\tau$. As an intuition, imagine that the sender commits to a signaling scheme such that each receiver's type best responds by playing a different action. Then, by observing the receiver's action, the sender gets to know the receiver's type with certainty. In general, for a given signaling scheme, there might be many different receiver's types that are better off playing the same action. In order to handle this problem and build the required estimates of the frequencies of the receiver's types, we use insights from the *bandit linear optimization* literature, and, in particular, we use the concept of *barycentric spanner* introduced by Awerbuch and Kleinberg [4].

For every block $I_\tau$ with $\tau \in [Z]$, we let $f_\tau : [0,1]^n \to \mathbb{R}$ be a function that, given a vector $\mathbf{x} = [x_1, \ldots, x_n] \in [0,1]^n$, returns the sum of the number of times the receiver's types in $\mathcal{K}$ were active during block $I_\tau$, weighted by the coefficients defined by the vector $\mathbf{x}$. Formally, the following definition holds:

$$f_\tau(\mathbf{x}) := \sum_{k \in \mathcal{K}} x_k \sum_{t \in B_\tau} \mathbb{I}\{k^t = k\},$$

where $\mathbb{I}\{k^t = k\}$ is an indicator function that is equal to 1 if and only if it is the case that $k^t = k$, while it is 0 otherwise. Notice that, for a given $\tau \in [Z]$ and $k \in \mathcal{K}$, the term $\sum_{t \in B_\tau} \mathbb{I}\{k^t = k\}$ is a constant, and, thus, the function $f_\tau$ is linear. Intuitively, $f_\tau$ is the key element that allows us to connect the utilities that we need to estimate with the actual quantities we can estimate through the use of barycentric spanners.

The first crucial step is to restrict the attention to posteriors that can be induced with at least some ('not too small') probability. This ensures that our estimators have a limited range. Given a probability threshold $\sigma \in (0,1)$, we denote with $\Xi^{\circledcirc} \subseteq \Xi$ the set of posteriors that can be induced with probability at least $\sigma$ by some signaling scheme. We can verify whether a given posterior $\boldsymbol{\xi} \in \Xi$ belongs to $\Xi^{\circledcirc}$ by solving an LP. Formally, $\boldsymbol{\xi} \in \Xi^{\circledcirc}$ if and only if the following set of linear equations admits a feasible solution $\mathbf{w} \in \Delta_\Xi$:

$$w_{\boldsymbol{\xi}} \geq \sigma \tag{10a}$$

$$\sum_{\boldsymbol{\xi} \in \Xi} w_{\boldsymbol{\xi}} \boldsymbol{\xi}_\theta = \mu_\theta \qquad\qquad \forall \theta \in \Theta. \tag{10b}$$

We define $\mathcal{R}$ as the set of all the tuples $\mathbf{a} = (a^k)_{k \in \mathcal{K}} \in \bigtimes_{k \in \mathcal{K}} \mathcal{A}$ for which there exists a posterior $\boldsymbol{\xi} \in \Xi^{\circledcirc}$ such that, for every receiver's type $k \in \mathcal{K}$, the action $a^k$ specified by the tuple is a best response to $\boldsymbol{\xi}$ for type $k$. Formally:

$$\mathcal{R} := \bigcup_{\boldsymbol{\xi} \in \Xi^{\circledcirc}} \left( b_{\boldsymbol{\xi}}^1, \ldots, b_{\boldsymbol{\xi}}^n \right),$$

where we recall that $b_{\boldsymbol{\xi}}^k$ denotes the best response of type $k \in \mathcal{K}$ under posterior $\boldsymbol{\xi}$. Intuitively, $\mathcal{R}$ is the set of tuples of receiver's best responses which result from the posteriors that the sender can induce with probability at least $\sigma$. [10]

Given a tuple $\mathbf{a} = (a^k)_{k \in \mathcal{K}} \in \mathcal{R}$ and a receiver's action $a \in \mathcal{A}$, we denote with $\mathbb{I}_{(\mathbf{a}=a)} \in \{0,1\}^n$ an indicator vector whose $k$-th component is equal to 1 if and only if type $k \in \mathcal{K}$ plays action $a$ in $\mathbf{a}$, *i.e.*, it holds $a^k = a$. Moreover, we define $\mathcal{X}$ as the set of all the indicators vectors; formally, $\mathcal{X} := \{\mathbb{I}_{(\mathbf{a}=a)} \mid \mathbf{a} \in \mathcal{R}, a \in \mathcal{A}\}$.

Since the set $\mathcal{X}$ is a finite (and hence compact) subset of the Euclidean space $\mathbb{R}^n$, we can use the following proposition due to Awerbuch and Kleinberg [4] to introduce the *barycentric spanner* of $\mathcal{X}$.

**Proposition 1** ([4], Proposition 2.2). *If $\mathcal{X}$ is a compact subset of an $n$-dimensional vector space $\mathcal{V}$, then there exists a set $\mathcal{H} = \{\mathbf{h}^1, ..., \mathbf{h}^n\} \subseteq \mathcal{X}$ such that for all $\mathbf{x} \in \mathcal{X}$, $\mathbf{x}$ may be expressed as a linear combination of elements of $\mathcal{H}$ using coefficients in $[-1,1]$. That is, for all $\mathbf{x} \in \mathcal{X}$, there exists a vector of coefficients $\boldsymbol{\lambda} = [\lambda_1, \ldots, \lambda_n] \in [-1,1]^n$ such that $\mathbf{x} = \sum_{i \in [n]} \lambda_i \mathbf{h}^i$. The set $\mathcal{H}$ is called* barycentric spanner *of $\mathcal{X}$.*

In the following, we denote with $\mathcal{H} := \{\mathbf{h}^1, ..., \mathbf{h}^n\} \subseteq \mathcal{X}$ a barycentric spanner of $\mathcal{X}$. Notice that, since each element $\mathbf{h} \in \mathcal{H}$ of the barycentric spanner belongs to $\mathcal{X}$ by definition, there exist a tuple $\mathbf{a} \in \mathcal{R}$ and a receiver's action $a \in \mathcal{A}$ such that $\mathbf{h}$ is equal to the indicator vector $\mathbb{I}_{(\mathbf{a}=a)}$. Moreover, by definition of $\mathcal{R}$, there exists a posterior $\boldsymbol{\xi} \in \Xi^{\circledcirc}$ such that the tuple of best responses $\left(b^1_{\boldsymbol{\xi}}, \ldots, b^n_{\boldsymbol{\xi}}\right)$ coincides with $\mathbf{a}$.

Next, we describe how Algorithm 1 computes the required estmates. During the exploration phase of block $I_\tau$ with $\tau \in [Z]$, one iteration is devoted to each element $\mathbf{h} \in \mathcal{H}$ of the barycentric spanner, so as to get an estimate of $f_\tau(\mathbf{h})$. During such iteration, the algorithm plays a signaling scheme $\mathbf{w} \in \Delta_\Xi$ that is feasible for the LP defined by Constraints (10) where the posterior $\boldsymbol{\xi} \in \Xi^{\circledcirc}$ is that associated to $\mathbf{h}$. As a result, the set of all such signaling schemes can be used as $W^{\circledcirc}$ in Algorithm 1. Moreover, when the induced receiver's posterior is $\boldsymbol{\xi}$ and the receiver responds by playing action $a$, the algorithm sets a variable $p_\tau(\mathbf{h})$ to the value $\frac{1}{w_{\boldsymbol{\xi}}}$, otherwise $p_\tau(\mathbf{h})$ is set to 0.

The following lemma shows that the variables $p_\tau(\mathbf{h})$ computed by the algorithm during each block $I_\tau$ with $\tau \in [Z]$ are unbiased estimates of the values $f_\tau(\mathbf{h})$.

**Lemma 4.** *For any $\tau \in [Z]$ and $\mathbf{h} \in \mathcal{H}$, it holds $\mathbb{E}\left[p_\tau(\mathbf{h}) \cdot |I_\tau|\right] = f_\tau(\mathbf{h})$.*

*Proof.* First, recall that $p_\tau(\mathbf{h}) = \frac{1}{w_{\boldsymbol{\xi}}}$ if and only if during the iteration of exploration devoted to $\mathbf{h}$, the induced receiver's posterior is $\boldsymbol{\xi}$ and she/he best responds by playing $a$ (otherwise, $p_\tau(\mathbf{h}) = 0$). Since the iteration is selected uniformly at random over the block $I_\tau$ and the sequence of receiver's types $\mathbf{k} = \{k^t\}_{t\in[T]}$ is chosen adversarially before the beginning of the game, we can conclude that also the receiver's type for that iteration is picked uniformly at random. Thus, $\mathbb{E}\left[p_\tau(\mathbf{h})\right] = \frac{1}{w_{\boldsymbol{\xi}}} \cdot w_{\boldsymbol{\xi}} \cdot \mathbb{P}\Big\{\text{randomly chosen type from } I_\tau \text{ best responds to } \boldsymbol{\xi} \text{ consistently with } \mathbf{h}\Big\}$, where by best responding consistently we mean that the type $k \in \mathcal{K}$ is such that $h_k = 1$, *i.e.*, she plays action $a$ in $\mathbf{a}$. By using the definition of $f_\tau(\mathbf{h})$, we can write the following:

$$\mathbb{E}\left[p_\tau(\mathbf{h})\right] = \frac{\sum_{k\in\mathcal{K}:h_k=1} f_\tau(\mathbf{e}^k)}{|I_\tau|} = \frac{f_\tau(\mathbf{h})}{|I_\tau|},$$

where $\mathbf{e}^k \in \mathbb{R}^n$ denotes an $n$-dimensional vector whose $k$-th component is 1, while others components are 0. $\square$

For any $\mathbf{x} \in \mathcal{X}$, we let $\boldsymbol{\lambda}(\mathbf{x}) = [\lambda_1(\mathbf{x}), \ldots, \lambda_n(\mathbf{x})] \in [-1, 1]^n$ be the vector of coefficients representing $\mathbf{x}$ with respect to basis $\mathcal{H}$. Formally, we can write $\mathbf{x} = \sum_{i\in[n]} \lambda_i(\mathbf{x})\mathbf{h}^i$.

For any posterior $\boldsymbol{\xi} \in \Xi^{\circledcirc}$, let $\mathbf{a}[\boldsymbol{\xi}] \in \mathcal{R}$ be such that $\mathbf{a}[\boldsymbol{\xi}] = \left(b^1_{\boldsymbol{\xi}}, \ldots, b^n_{\boldsymbol{\xi}}\right)$. Then, for each $\tau \in [Z]$, let us define

$$\tilde{u}^{\mathsf{s}}_{I_\tau}(\boldsymbol{\xi}) := \sum_{a\in\mathcal{A}} \sum_{k\in\mathcal{K}} \lambda_k\left(\mathbb{I}_{\mathbf{a}[\boldsymbol{\xi}]=a}\right) p_\tau\left(\mathbf{h}^k\right) \sum_{\theta\in\Theta} \xi_\theta u^{\mathsf{s}}_\theta(a).$$

Letting $u^{\mathsf{s}}_{I_\tau}(\boldsymbol{\xi}) := \frac{1}{|I_\tau|} \sum_{t\in\tau} u^{\mathsf{s}}(\boldsymbol{\xi}, k^t)$ be the sender's average utility achieved by inducing the receiver's posterior $\boldsymbol{\xi} \in \Xi^{\circledcirc}$ during each iteration of block $I_\tau$ with $\tau \in [Z]$, the following lemma shows that $\tilde{u}^{\mathsf{s}}_{I_\tau}(\boldsymbol{\xi})$ is an unbiased estimator of $u^{\mathsf{s}}_{I_\tau}(\boldsymbol{\xi})$, and, additionally, the range in which the estimator values lie is not to large.

**Lemma 5.** *For any posterior $\boldsymbol{\xi} \in \Xi^{\circledcirc}$ and $\tau \in [Z]$, it holds $\mathbb{E}\left[\tilde{u}^{\mathsf{s}}_{I_\tau}(\boldsymbol{\xi})\right] = u^{\mathsf{s}}_{I_\tau}(\boldsymbol{\xi})$. Moreover, $\tilde{u}^{\mathsf{s}}_{I_\tau}(\boldsymbol{\xi}) \in [-\frac{mn}{\sigma}, \frac{mn}{\sigma}]$.*

*Proof.* The first statement follows from the following relations:

$$\mathbb{E}\left[\tilde{u}^{\mathsf{s}}_{I_\tau}(\boldsymbol{\xi})\right] = \mathbb{E}\left[\sum_{a\in\mathcal{A}} \sum_{k\in\mathcal{K}} \lambda_k\left(\mathbb{I}_{\mathbf{a}[\boldsymbol{\xi}]=a}\right) p_\tau\left(\mathbf{h}^k\right) \sum_{\theta\in\Theta} \xi_\theta u^{\mathsf{s}}_\theta(a)\right]$$

$$= \sum_{a\in\mathcal{A}} \sum_{k\in\mathcal{K}} \lambda_k\left(\mathbb{I}_{\mathbf{a}[\boldsymbol{\xi}]=a}\right) \mathbb{E}\left[p_\tau\left(\mathbf{h}^k\right)\right] \sum_{\theta\in\Theta} \xi_\theta u^{\mathsf{s}}_\theta(a)$$

$$= \sum_{a\in\mathcal{A}} \sum_{\theta\in\Theta} \xi_\theta u^{\mathsf{s}}_\theta(a) \sum_{k\in\mathcal{K}} \lambda_k\left(\mathbb{I}_{\mathbf{a}[\boldsymbol{\xi}]=a}\right) \mathbb{E}\left[p_\tau\left(\mathbf{h}^k\right)\right]$$

$$= \sum_{a\in\mathcal{A}} \sum_{\theta\in\Theta} \xi_\theta u^{\mathsf{s}}_\theta(a) \sum_{k\in\mathcal{K}} \lambda_k\left(\mathbb{I}_{\mathbf{a}[\boldsymbol{\xi}]=a}\right) \frac{f_\tau\left(\mathbf{h}^k\right)}{|I_\tau|} \qquad\qquad \text{(By Lemma 4)}$$

$$= \sum_{a\in\mathcal{A}} \sum_{\theta\in\Theta} \xi_\theta u^{\mathsf{s}}_\theta(a) \sum_{k\in\mathcal{K}} \frac{f_\tau\left(\mathbb{I}_{\mathbf{a}[\boldsymbol{\xi}]=a}\right)}{|I_\tau|} \qquad\qquad \text{(By definition of } f_\tau)$$

$$= u^{\mathsf{s}}_{I_\tau}(\boldsymbol{\xi}),$$

where the last equality holds by using again the definition of $f_\tau$ and re-arranging the terms.

As for the second statement, since $\lambda_k\left(\mathbb{I}_{\mathbf{a}[\boldsymbol{\xi}]=a}\right) \in [-1, 1]$, $\sum_{\theta \in \Theta} \xi_\theta u_\theta^{\mathsf{s}}(a) \in [0, 1]$, and $p_\tau\left(\mathbf{h}^k\right) \in \left[0, \frac{1}{\sigma}\right]$, it is easy to show that $\tilde{u}_{I_\tau}^{\mathsf{s}}(\boldsymbol{\xi}) \in [-\frac{mn}{\sigma}, \frac{mn}{\sigma}]$. $\qquad\square$

In the next lemma, we show that there always exists a best-in-hindsight signaling scheme that uses (*i.e.*, induces with positive probability) only a small number of posteriors. This is the final step needed to show that the estimators $\tilde{u}_{I_\tau}^{\mathsf{s}}(\boldsymbol{\xi})$ allow to compute slightly biased estimates of the utilities needed by the full-information algorithm.

**Lemma 6.** *Given a sequence of receiver's types* $\mathbf{k} = \{k^t\}_{t \in [T]}$, *there always exists a best-in-hindsight signaling scheme* $\mathbf{w}^\star \in W^\star$ *such that the set of posteriors it induces with positive probability* $\left\{\boldsymbol{\xi} \in \Xi \mid w_{\boldsymbol{\xi}}^\star > 0\right\}$ *has cardinality at most the number of states* $d$.

*Proof.* Notice that a best-in-hindsight signaling scheme $\mathbf{w}^\star \in W^\star$ can be computed by solving the following LP:

$$\max_{\mathbf{w} \in \Delta_\Xi} \quad \sum_{t \in [T]} \sum_{\boldsymbol{\xi} \in \Xi} w_{\boldsymbol{\xi}} u^{\mathsf{s}}(\mathbf{w}, k^t)$$

$$\text{s.t.} \quad \sum_{\boldsymbol{\xi} \in \Xi} w_{\boldsymbol{\xi}} \xi_\theta = \mu_\theta \qquad\qquad \forall \theta \in \Theta.$$

Since the LP has $d$ equalities, it always admits an optimal basic feasible solution in which at most $d$ variables $w_{\boldsymbol{\xi}}$ are greater than 0. This concludes the proof. $\qquad\square$

Then, we define the $W^\circ$ used by Algorithm 1 as the set of signaling schemes $\mathbf{w} \in W^\star$ whose support is at most $d$, *i.e.*, it is the case that $|\{\boldsymbol{\xi} \in \Xi \mid w_{\boldsymbol{\xi}} > 0\}| \le d$. By definition of $W^\star$ and Lemma 6, it is easy to see that a best-in-hindsight signaling scheme is always guaranteed to be in the set $W^\circ$.

Letting $\tilde{u}_{I_\tau}^{\mathsf{s}}(\mathbf{w}) := \sum_{\boldsymbol{\xi} \in \Xi^\circ} w_{\boldsymbol{\xi}} \tilde{u}_{I_\tau}^{\mathsf{s}}(\boldsymbol{\xi})$ for every $\mathbf{w} \in W^\circ$ and $\tau \in [Z]$, the following lemma shows that each $\tilde{u}_{I_\tau}^{\mathsf{s}}(\mathbf{w})$ is a biased estimator of the sender's average utility $u_{I_\tau}^{\mathsf{s}}(\mathbf{w})$ in block $I_\tau$, while also providing bounds on the bias and the range of the estimators. This final result allows us to effectively use the estimators $\tilde{u}_{I_\tau}^{\mathsf{s}}(\mathbf{w})$ defined above in Algorithm 1.

**Lemma 7.** *For any signaling scheme* $\mathbf{w} \in W^\circ$ *and* $\tau \in [Z]$, *it holds* $u_{I_\tau}^{\mathsf{s}}(\mathbf{w}) \ge \mathbb{E}\left[\tilde{u}_{I_\tau}^{\mathsf{s}}(\mathbf{w})\right] \ge u_{I_\tau}^{\mathsf{s}}(\mathbf{w}) - d\sigma$. *Moreover, it is the case that* $\tilde{u}_{I_\tau}^{\mathsf{s}}(\mathbf{w}) \in [-\frac{mn}{\sigma}, \frac{mn}{\sigma}]$.

*Proof.* By using Lemma 5, it is easy to check that the left inequality in the first statement holds:

$$u_{I_\tau}^{\mathsf{s}}(\mathbf{w}) = \sum_{\boldsymbol{\xi} \in \Xi} w_{\boldsymbol{\xi}} u_{I_\tau}^{\mathsf{s}}(\boldsymbol{\xi}) \ge \sum_{\boldsymbol{\xi} \in \Xi^\circ} w_{\boldsymbol{\xi}} u_{I_\tau}^{\mathsf{s}}(\boldsymbol{\xi}) = \sum_{\boldsymbol{\xi} \in \Xi^\circ} w_{\boldsymbol{\xi}} \mathbb{E}\left[\tilde{u}_{I_\tau}^{\mathsf{s}}(\boldsymbol{\xi})\right] = \mathbb{E}\left[\tilde{u}_{I_\tau}^{\mathsf{s}}(\mathbf{w})\right].$$

Moreover, it is the case that:

$$\mathbb{E}\left[\tilde{u}_{I_\tau}^{\mathsf{s}}(\mathbf{w})\right] = \sum_{\boldsymbol{\xi} \in \Xi^\circ} w_{\boldsymbol{\xi}} \mathbb{E}\left[\tilde{u}_{I_\tau}^{\mathsf{s}}(\boldsymbol{\xi})\right]$$

$$= \sum_{\boldsymbol{\xi} \in \Xi^\circ} w_{\boldsymbol{\xi}} u_{I_\tau}^{\mathsf{s}}(\boldsymbol{\xi}) \qquad\qquad \text{(By Lemma 5)}$$

$$= u_{I_\tau}^{\mathsf{s}}(\mathbf{w}) - \sum_{\boldsymbol{\xi} \in \Xi \setminus \Xi^\circ} w_{\boldsymbol{\xi}} u_{I_\tau}^{\mathsf{s}}(\boldsymbol{\xi}) \qquad\qquad \text{(By definition of } u_{I_\tau}^{\mathsf{s}}(\mathbf{w}))$$

$$\ge u_{I_\tau}^{\mathsf{s}}(\mathbf{w}) - \sum_{\boldsymbol{\xi} \in \Xi \setminus \Xi^\circ} w_{\boldsymbol{\xi}} \qquad\qquad \text{(Since } u_{I_\tau}^{\mathsf{s}}(\mathbf{w}) \le 1)$$

$$\ge u_{I_\tau}^{\mathsf{s}}(\mathbf{w}) - \sum_{\boldsymbol{\xi} \in \Xi \setminus \Xi^\circ} \sigma \qquad\qquad \text{(By definition of } \Xi^\circ \text{, it must be } w_{\boldsymbol{\xi}} < \sigma)$$

$$\ge u_{I_\tau}^{\mathsf{s}}(\mathbf{w}) - d\sigma \qquad\qquad \text{(Since } \mathbf{w} \in W^\circ)$$

Finally, $\tilde{u}_{I_\tau}^{\mathsf{s}}(\mathbf{w}) \in [-\frac{mn}{\sigma}, \frac{mn}{\sigma}]$ follows from the fact that, by definition, $\tilde{u}_{I_\tau}^{\mathsf{s}}(\mathbf{w})$ is the weighted sum of quantities within the range $[-\frac{mn}{\sigma}, \frac{mn}{\sigma}]$, with the weights sum being at most 1. $\qquad\square$

### D.3 Proof of Theorem 4

**Theorem 4.** *Given an online Bayesian persuasion problem with partial feedback, there exist $W^\circ \subseteq W^\star$, $W^\circledcirc \subseteq W^\star$, and estimators $\tilde{u}^{\mathsf{s}}_{I_\tau}(\mathbf{w})$ such that Algorithm 1 provides the following regret bound:*

$$R^T \le O\left(\frac{nm^{2/3}d\log^{1/3}(mn+d)}{T^{1/5}}\right).$$

*Proof.* By setting $\sigma := d^{-2/5}T^{-1/5}$, it is sufficient to run Algorithm 1 with estimators $u^{\mathsf{s}}_{I_\tau}(\mathbf{w})$ for every $\mathbf{w} \in W^\circ$ computed as previously described in this section. Thus, it holds $|W^\circledcirc| = n$ and $\eta = mnd^{2/5}T^{1/5}$. By Theorem 3, the following holds:

$$
\begin{aligned}
R^T &\le O\left(\frac{|W^\circledcirc|^{1/3}\eta^{2/3}log^{1/3}|W^\circ|}{T^{1/3}}\right) + O(\iota) \\
&= O\left(\frac{n^{1/3}\left(mnd^{2/5}T^{1/5}\right)^{2/3}\log^{1/3}|W^\circ|}{T^{1/3}}\right) + O\left(\frac{d}{d^{2/5}T^{1/5}}\right) \\
&= O\left(\frac{nm^{2/3}d^{4/15}\left(d\log\left(m^2n+d\right)\right)^{1/3}}{T^{1/5}}\right) + O\left(\frac{d^{3/5}}{T^{1/5}}\right) \\
&= O\left(\frac{nm^{2/3}d^{3/5}\log^{1/3}(mn+d)}{T^{1/5}}\right).
\end{aligned}
$$

This concludes the proof. $\qquad\square$

## Footnotes

[9]The polytopes were computed using `Polymake`, a tool for computational polyhedral geometry [3, 20].

[10] Let us remark that the sets $\Xi^{\circledcirc}$ and $\mathcal{R}$ depend on the given threshold $\sigma \in (0,1)$. In the following, for the ease of notation, we omit such dependence, as the actual value of $\sigma$ that the two sets refer to will be clear from context.