[Reviews · NeurIPS 2020]

Review 1

Summary and Contributions: Considers Bayesian persuasion in an online setting where receiver types are unknown. First shows the result (surprising to me) that, even given a known distribution over receiver types, computing an approximately optimal signaling scheme is NP-hard. This translates to a hardness for online algorithms. Then gives (inefficient) algorithms for the online problem. First shows that it suffices to try to induce a finite set of posteriors W* (exponential-sized in the number of states of nature). Thus, a no-regret algorithm like polynomial weights works. Finally, considers the partial information setting, which is much more complex. The approach proposed is to mix some explore and exploit within an epoch, then give feedback from the epoch to the full-information algorithm, and repeat. The analysis is quite involved and only summarized in the main text.

Strengths: I think the paper studies a very interesting, novel problem and gives interesting results. It is clear and well-written. It is well-referenced and placed within the broader literature. The results are intriguing.

Weaknesses: This is hard to avoid, but many technical details were left to the appendix.

Correctness: Unfortunately I did not check the proofs, however, the high-level overview makes sense and I have reasonable confidence in the results based on the main text.

Clarity: Yes.

Relation to Prior Work: Yes.

Reproducibility: Yes

Additional Feedback: Nice job, I enjoyed the paper. For both hardness results and upper bounds, think it would be useful to emphasize which parameter the algorithm is exponential in (I understood it to be just "d"). This is very interesting. I was also very interested in the NP-hardness result and would have enjoyed some more intuition for it in the main text. -------- After author response: Thanks for the response. During discussion, I agreed with other reviewers about some downsides, but felt they were not severe and still felt very positive about the paper.


Review 2

Summary and Contributions: This paper builds on the extremely impactful model of signal sending called Bayesian Persuasion, from economics, by formulating a learning problem whose base is BP. There is a sender and receiver of signals, and the sender wants to design a signaling scheme to optimize his payoff based on the action the receiver takes. This paper puts this into a repeated framework and asks how a sender can learn a signaling scheme over the repeated rounds that does well relative to the best fixed signaling scheme. Here, this is not just the offline optimal, because the receiver each round has a type that affects her payoffs, and the sequence of types may be adversarial (but not adaptive). The authors give several results. First they show that computing the offline optimal is computationally hard, even to approximate. Then, relaxing the requirement of polynomial time, they give an algorithm for the full feedback setting, and show that standard no-regret algorithms can be used here. Then they use the full information algorithm as a subroutine to get a no-regret algorithm for the partial feedback case. --Update -- Thanks to the authors for your response. Re notation, I see the issue about many more symbols; one approach that I have found useful if you want to use the same letter is to use a little text (e.g. W_{OPT}, W_{Greedy}, etc) that carries the information in the symbol itself. In any case, my score is unchanged.

Strengths: The model is a very well known and important one, and the paper is technically very interesting. These results appear to be novel to me and correct, and the paradigm is very powerful so it’s conceivable that the algorithms described could be applied in practice.

Weaknesses: I think between the notation and maybe drawing things out more than necessary, the clarity of the paper suffers. For instance, showing that any optimal scheme could be written as a combination of a finite number of signals is pretty intuitive (since all that ultimately matters across signaling schemes is the distribution of actions taken over the finite action set), so I would move this to the appendix and make a more informal discussion that is easier for threader to follow. Notationally, the symbols are too similar for me to keep easy track of - i.e. there’s W, W circle, W*, W hat, W with concentric circles… Maybe if you used different letters, it would be hard for the reason that you would convey less information with the notation, so I won’t say for certain that there is a better option, but the current state imposes a lot of attention cost on the reader, in my opinion.

Correctness: As far as I can tell, the main results appear correct, though I am not a complexity person so I was not able to fully verify that result.

Clarity: See weaknesses above - there are notational problems and more density than appears necessary. With enough time and attention, it is fairly well-written, but I think the writing could be improved.

Relation to Prior Work: Mostly, but - while I am no expert in this literature, there do appear to be some BP models focused specifically on algorithmic aspects e.g. Dughmi and Xu, and Babichenko and Barman. These particular examples appear to have different setups, and so it’s not surprising that they seem have different answers to the question of complexity of computing approximately optimal signaling schemes, but I would still prefer to have a clear comparison to these models. I don’t know the literature too well so I’m not claiming these are the best or only examples, but just that it appears there is a computational BP literature out there not discussed in the related work.

Reproducibility: Yes

Additional Feedback:


Review 3

Summary and Contributions: This paper studies online Bayesian persuasion, in which the sender needs to persuade a sequence of receivers with adversarially chosen types from a finite set of possible types. The paper first shows that even the offline problem is NP-hard. Without time constraints, the paper then presents algorithms with a good regret bound in the full information setting. They also extend this result to the partial information feedback setting via a reduction. --After feedback-- Thanks for the feedback. Here is a minor comment. For the response "thus showing that the number of states is the only source of hardness", I don't fully agree with this sentence as the lower bound is worst-case. There could be other natural constraints that make the problem tractable. But I can see your point.

Strengths: The paper initiates the framework of online Bayesian persuasion. The algorithm is mainly based on the characterization shown in Lemma 1: the sender only needs to consider a finite set of signal schemes.

Weaknesses: The negative result of the paper shows that the problem is NP-hard. The authors take it as a permission to use algorithms with bad running time. I think it’s more interesting to take the other direction: getting poly-time algorithms in some more restricted settings.

Correctness: The results in the paper look correct to me.

Clarity: The paper’s presentation is clear.

Relation to Prior Work: The paper clearly discusses the relation to prior work. Here is a paper that might be related: “Incentivizing Exploration with Heterogeneous Agents”. It studies how to persuade agents of different types to do exploration.

Reproducibility: Yes

Additional Feedback: Line 41: Technically, your result only says the problem is NP-hard. It might be more accurate to say in that way. Otherwise it sounds like you have proved P!=NP.

[Author Response · NeurIPS 2020]

**Reviewer 1:** Thanks for your feedback! Your understanding is correct: the algorithm is exponential in the number of states of nature $d$. The algorithm is tractable in all the other parameters. We will emphasize this aspect in the final version of the paper. If the paper is accepted, we will provide more intuition on the NP-hardness proof using the ninth content page for the camera ready version.

**Reviewer 2:** Thanks for your comments!
- Re "*showing that any optimal signaling scheme can be written as a combination of a finite number of signals*". We agree with the reviewer that the proof of this results is intuitive. We decided to include this result in the main body of the paper since it is crucial when proving that the running time of our algorithm is exponential in the number of states, and polynomial in the other parameters. We will include a more informal discussion of the results using the ninth content page for the camera ready version.
- Re "*notation*". Thanks for your feedback on the notation. We will try to simplify the notation as much as possible. However, a formal presentation of our technical results will inevitably require many symbols. On the use of the symbol $W$: we preferred to use different variants of the same symbol as they denote entities which are conceptually related.
- Re "*related works*". Due to the limited space we presented only the results closest to ours. We agree with the reviewer that the algorithmic Bayesian persuasion literature includes many other works that adopt models technically different from ours. We will include some additional discussion on these alternative frameworks in the appendix.

**Reviewer 3:** Thanks for your feedback. Our hardness results show that when the number of states of nature is free the problem is not approximable. However, when the number of states of nature $d$ is fixed, the algorithm requires polynomial time, thus showing that the number of states is the only source of hardness. Indeed, this may be considered a first restricted scenario where our algorithm has a poly-time guarantee. To the best of our knowledge this is the first work studying Bayesian persuasion in the online setting. Hence, the first results that need to be provided are arguably the hardness and fixed-parameter tractability of the problem in the general setting, which is precisely what we present in the paper. The study of ad-hoc algorithms for other restricted settings would not be well motivated without the set of general results that we are presenting. We agree with the reviewer that the study of poly-time algorithms for restricted settings is a natural next step for this line of research.

We will add the suggested paper to the related works (thanks for the pointer!) and reword line 41.

[Meta-Review · NeurIPS 2020]

Three knowledgeable referees all agreed the paper should be accepted (in both their initial reviews and after reviewing the authors' response). The paper contributes interesting complexity results about Bayesian Persuasion (it is NP-hard in general), and give algorithms for solving the problem. I agree that the paper makes a nice contribution to the NeurIPS community, and therefore recommend it be accepted.